# Diffusion Transportation Cost for Domain Adaptation

## Abstract

In recent years, there has been considerable interest in leveraging the Optimal Transport (OT) problem for domain adaptation, a strategy shown to be highly effective. However, a less explored aspect is the choice of the transportation cost function, as most existing methods rely on the pairwise squared Euclidean distances for the transportation cost, potentially overlooking important intra-domain geometries. This paper presents Diffusion-OT, a new transport cost for the OT problem, designed specifically for domain adaptation. By utilizing concepts and tools from the field of manifold learning, specifically diffusion geometry, we derive an operator that accounts for the intra-domain relationships, thereby extending beyond the conventional inter-domain distances. This operator, which quantifies the probability of transporting between source and target samples, forms the basis for our transportation cost. We provide proof that the proposed operator is in fact a diffusion operator, demonstrating that the cost function is defined by an anisotropic diffusion process between the domains. In addition, to enhance performance, we integrate source labels into the operator, thereby guiding the anisotropic diffusion according to the classes. We showcase the effectiveness of Diffusion-OT through comprehensive experiments, demonstrating its superior performance compared to recent methods across various benchmarks and datasets.

## 1 Introduction

Unsupervised Domain Adaptation (UDA) is a widely explored field within machine learning. In UDA, given labeled data from a domain denoted as the source domain, we aim to train a model that will generalize well to unlabeled data from a different domain denoted as the target domain. Optimal Transport (OT) stands out as a key method for addressing the UDA problem. The essence of the OT problem lies in finding the most efficient way to transport from one distribution to another while minimizing a transportation cost. It offers a robust approach to measuring the similarity between probability density functions and facilitating an optimal mapping between them. This capability makes it well-suited for UDA tasks, where there is a need for distribution alignment.

An established approach to utilizing OT for UDA is the Optimal Transport for Domain Adaptation (OTDA) framework (Courty et al., 2016), which directly employs the transport plan to map source samples to the target domain via barycentric mapping. While OTDA has proven effective in addressing the UDA task, the OT solution may not always yield the optimal mapping for maximizing target accuracy. To improve accuracy, a common practice is to add a regularization term into the OT problem formulation (Courty et al., 2014; 2016; Flamary et al., 2014). Alternatively, in recent years, the OT problem has been incorporated into deep learning models. One notable method, termed DeepJDOT (Damodaran et al., 2018), uses the OT problem to address the discrepancy between source and target distributions, subsequently leveraging the obtained transportation plan to train a feature extractor implemented through a deep model. Following DeepJDOT, numerous studies have proposed OT-based deep models, aimed at improving target accuracy in the UDA task. Some have suggested different OT formulations to be used within the DeepJDOT framework (Fatras et al., 2021; Nguyen et al., 2022a), while others have introduced entirely new architectures and objective functions (Chen et al., 2018; Lee et al., 2019; Balaji et al., 2020). While research on utilizing OT for DA tasks is extensive and constantly evolving, one aspect that remains relatively unexplored is the selection of the transportation cost function. The transportation cost can take the form of any dissimilarity measure between samples in the source and target domains. Nevertheless, the predominant choice tends to

be a distance function, with the most common choice being the squared Euclidean distance, computed between source and target samples. Only a small number of papers have suggested using a cost function diverging from the typical Euclidean distance measure (Tai et al., 2021; Nguyen et al., 2022b; Duque et al., 2023). Notably, this exploration of the transportation cost function is even rarer in the DA context, with only a few studies proposing alternative approaches, such as employing a weighted Euclidean distance as the transport cost (Li et al., 2020; Xu et al., 2020).

In this paper, we depart from reliance on distance functions as the transportation cost, and instead, focus on the probability of transportation from source samples to target samples. To achieve this, we introduce a new transportation cost, which we term Diffusion-OT, based on an operator composed of three diffusion operators. The construction of the operator uses concepts and tools from the field of manifold learning, specifically, diffusion geometry (Coifman & Lafon, 2006). Initially, we construct a graph solely from the source samples, where each node represents a sample in the source data. From this graph, we build the first diffusion operator, which aims at preserving the local geometry of the source domain. When source labels are available, they are utilized to further guide the diffusion process, ensuring that each source sample can only diffuse (transport) to other samples within the same label category. Subsequently, we build a bipartite graph incorporating both source and target samples to capture the inter-domain relationships. We construct the second operator from this graph, which facilitates cross-domain diffusion. For the third diffusion operator, we construct a graph exclusively from target samples, aimed at capturing intra-domain similarities within the target domain. Taking the product of the three aforementioned operators forms a composite operator. We show theoretically that this composite operator is a diffusion operator, demonstrating an anisotropic diffusion process between the source and target domains. Our analysis underscores that this process is primarily influenced by the intrinsic geometry of the domains and their discrepancy conveyed by the Radon-Nikodym (RN) derivative.

**Our main contribution.** We introduce a new transportation cost, termed Diffusion-OT, which is based on a composite diffusion operator consisting of three diffusion steps: (i) in the source domain, (ii) across domains, and (iii) in the target domain. This framework enables the learning of the geometries and relationships both between and within the two domains, distinguishing it from conventional approaches by considering both inter-domain distances and intra-domain structures. By incorporating source label information into our cost, we eliminate the need for regularization terms, ensuring compatibility with any OT solver and problem formulation, as we demonstrate in Section 5.2. Additionally, the proposed cost is straightforward to compute and is derived directly from the data, in contrast to competing cost functions that necessitate learning the cost and are thus limited to deep learning frameworks. Finally, we present experiments that demonstrate the effectiveness of our method compared to competing approaches. Specifically, by incorporating Diffusion-OT into a deep domain adaptation framework, we attain superior performance on benchmark datasets compared to the baseline and other OT-based methods. Additionally, we evaluate our cost on benchmarks of Motor Imagery (MI), showcasing state-of-the-art (SOTA) results on non-Euclidean data.

## 2 RELATED WORK

UDA is a subfield of machine learning that addresses the challenges of domain shifts and generalization, where a model trained on data from a source domain performs poorly when applied to a target domain with a different data distribution. Various methods and metrics have been proposed to align the distributions of source and target domains, including Maximum Mean Discrepancy (MMD) (Gretton et al., 2012; Pan et al., 2010), Correlation Alignment (CORAL) (Sun et al., 2017), and Domain Adversarial Neural Networks (DANN) (Ganin et al., 2016), to mention just a few.

The Optimal Transport for Domain Adaptation (OTDA) framework was introduced by Courty et al. (2016). In addition to the general framework, the authors propose the integration of several regularization terms to improve performance. These include a class-based regularization term (Courty et al., 2014) and Laplace regularization (Flamary et al., 2014). A notable work is the JDOT framework introduced in Courty et al. (2017), which presents an optimization problem designed to simultaneously optimize both the transportation plan and the classifier. Building upon this concept, the DeepJDOT framework (Damodaran et al., 2018) further extends the idea by leveraging deep learning algorithms. In Fatras et al. (2021), the authors proposed enhancing the DeepJDOT framework by employing Unbalanced Optimal Transport (UOT), yielding improved results. A related work by

Nguyen et al. (2022a) suggested solving it using Partial Optimal Transport (POT) and presented a two-stage implementation for the DeepJDOT framework, further enhancing target accuracy, particularly when coupled with the POT formulation. The OT problem was later incorporated into various frameworks utilizing deep models and adversarial training (Chen et al., 2018; Lee et al., 2019; Xie et al., 2019; Balaji et al., 2020), all with the squared Euclidean distance serving as the ground cost.

Few papers in the literature have proposed alternative cost functions diverging from the standard distance metric. For instance, Tai et al. (2021) introduced an innovative approach termed SLA, which utilizes the OT formulation as a label assignment method. Gu et al. (2022) introduced a semi-supervised cost for Heterogeneous DA, with a similar approach proposed in Duque et al. (2023) for manifold alignment. Asadulaev et al. (2022) proposed a neural network-based framework for learning task-specific semi-supervised cost functions. Although they do not specifically propose a transport cost, Yan et al. (2018) and Xu et al. (2019) present innovative approaches to incorporate both intra-domain and inter-domain relationships within Gromov-Wasserstein-based frameworks.

In the context of UDA, Li et al. (2020) introduced ETD, which employs a weighted Euclidean distance as the cost function, with weights estimated via an attention mechanism. Similarly, Xu et al. (2020) proposed RWOT, where the weights are dynamically computed using both spatial prototypical information and the pseudo-classification probability of target samples. Both methods have fundamental drawbacks compared to our proposed approach. First, the weights in ETD and RWOT are learned, rather than directly derived from the data as they are in Diffusion-OT, which limits their applicability to deep learning frameworks. Furthermore, the costs in ETD and RWOT, including the learned weights, consider only the inter-domain relationships and source label information, whereas our cost also incorporates intra-domain relationships for both source and target domains.

By integrating principles from diffusion maps (Coifman & Lafon, 2006) into our approach, we effectively capture intra-domain relationships and the underlying geometric structure of the data within our proposed cost. Diffusion maps, initially introduced as a dimensionality reduction technique, provide a method to analyze and visualize high-dimensional data. The core concept involves constructing a transition kernel, where transition probabilities are defined by the local similarities between data points. Diffusion geometry has since been widely applied in various fields, including data analysis (Coifman & Lafon, 2006), computer vision (Bronstein et al., 2010; Liu et al., 2012), and medical imaging (Haghverdi et al., 2015; Zheludev et al., 2015), among others.

For a more comprehensive review of the related work and diffusion geometry specifically, see Appendices A and B, respectively.

## 3 OPTIMAL TRANSPORT FOR DOMAIN ADAPTATION

We formulate the problem we consider and briefly describe the OTDA framework proposed in Courty et al. (2016).

**Problem formulation.** Let $\mathcal{X}$ be an input space and $\mathcal{Y}$ be a label space. We define a domain as a distribution $\mathcal{D}$ over the input space $\mathcal{X}$. Consider $N_s$ samples $\{x_i^s \in \mathcal{X}\}_{i=1}^{N_s}$ drawn i.i.d from a source domain $\mathcal{D}_s$, associated with labels $\{y_i^s \in \mathcal{Y}\}_{i=1}^{N_s}$, and $N_t$ unlabeled samples $\{x_j^t \in \mathcal{X}\}_{j=1}^{N_t}$, drawn i.i.d from a target domain $\mathcal{D}_t$. In UDA, the goal is to enable a predictive model $f$, trained on labeled source samples, to generalize well to target samples. There are two dominant approaches for UDA. In the first approach, a mapping function $g : \mathcal{X} \rightarrow \mathcal{X}$ is learned to align the source domain to the target domain, e.g., by minimizing the divergence between them. A predictive model $f$ is then trained using the mapped source samples, denoted by $z_i^s = g(x_i^s)$. In the second approach, we learn an embedding function $g : \mathcal{X} \rightarrow \mathcal{Z}$ that maps both domains to a latent space $\mathcal{Z}$ of domain-invariant features. Thus, a predictive model $f : \mathcal{Z} \rightarrow \mathcal{Y}$, trained using the source embeddings $\{z_i^s = g(x_i^s)\}_{i=1}^{N_s}$ with labels $\{y_i^s\}_{i=1}^{N_s}$ will generalize well on the target embeddings $\{z_j^t = g(x_j^t)\}_{j=1}^{N_t}$.

**OTDA framework.** We start by outlining the discrete OT formulation, which is consistently employed throughout this paper. For any source sample $x_i^s$ and target sample $x_j^t$, let $c(x_i^s, x_j^t)$ denote the cost to move a probability mass from $x_i^s$ to $x_j^t$, which is typically chosen to be the squared Euclidean distance. Let $\mathbf{C} \in \mathbb{R}^{N_s \times N_t}$ be the cost matrix, whose elements are $\mathbf{C}_{ij} = c(x_i^s, x_j^t)$. The discrete OT formulation is given by $\gamma^* = \underset{\gamma \in \Gamma}{\arg\min} \langle \gamma, \mathbf{C} \rangle_F$, where $\Gamma = \left\{ \gamma \in \mathbb{R}^{N_s \times N_t} | \gamma \mathbb{1}_{N_t} = \mathbf{a}, \gamma^T \mathbb{1}_{N_s} = \mathbf{b} \right\}$

denotes the set of transport plans that satisfy conservation of mass, $\mathbf{a} \in \mathbb{R}^{N_s}$ and $\mathbf{b} \in \mathbb{R}^{N_t}$ are the empirical distributions of the source and target samples, respectively, and $\langle \cdot, \cdot \rangle_F$ denotes the standard Frobenius inner product.

This optimization problem has been studied for many years and addressed through various algorithms. To date, one of the most commonly employed algorithms is the efficient Sinkhorn algorithm (Cuturi, 2013), which incorporates entropy regularization into the objective function. This leads to the following modified problem that can be solved iteratively using linear projections, offering reduced complexity:

$$\gamma^* = \arg\min_{\gamma \in \Gamma} \langle \gamma, \mathbf{C} \rangle_F + \lambda \Omega_s(\gamma), \tag{1}$$

where $\Omega_s(\gamma) = \sum_{i,j} \gamma(i,j) \log \gamma(i,j)$ is the negative entropy of $\gamma$, and $\lambda$ is a hyperparameter.

The OT problem can be utilized for UDA in both of the aforementioned approaches. Recent deep learning-based methods, such as Damodaran et al. (2018); Chen et al. (2018); Lee et al. (2019), incorporate OT into the loss function to learn domain-invariant representations, thereby following the second approach for UDA. In contrast, the OTDA framework follows the first approach, where, after solving the OT problem, source samples are transported to the target domain using the obtained transport plan. The transportation of any source sample $x_i^s$ is given by the barycentric mapping (Courty et al., 2016; Perrot et al., 2016), defined by: $z_i^s = \arg\min_{x \in \mathcal{X}} \sum_{j=1}^{N_t} \gamma_{ij}^* c(x, x_j^t)$.

## 4  DIFFUSION OPTIMAL TRANSPORT

In this work, we present Diffusion-OT, a new cost function for the OT problem, designed specifically for the UDA setting described in Section 3. Our premise is the manifold assumption, which is widely spread in modern high-dimensional data analysis and gives rise to an extensive body of work termed manifold learning (Roweis & Saul, 2000; Tenenbaum et al., 2000; Belkin & Niyogi, 2003; Coifman & Lafon, 2006). In the UDA setting, it implies that both domains are supported on the same hidden manifold $\mathcal{M}$ embedded in $\mathcal{X}$. When the squared Euclidean distance is used as the transport cost, whether explicitly stated or not, it is assumed that the source and target domains are embedded in $\mathbb{R}^d$. The manifold assumption extends this concept, allowing for the possibility that the samples may not lie within a Euclidean space. Building on this foundation, we propose defining the OT cost matrix $\mathbf{C}$ based on a diffusion operator, composed of three diffusion steps. Notably, the construction of this operator stems from diffusion geometry (Coifman & Lafon, 2006), which is a prevalent manifold learning approach (see Appendix B for details).

Initially, we build an operator solely from the source samples, aimed at capturing the local geometry within the source domain. We construct a graph $G_s$ with $N_s$ nodes, where each node represents a source sample. Assuming that the hidden manifold $\mathcal{M}$ is locally Euclidean, we use the Euclidean distance to build the affinity matrix of the graph, $\mathbf{K}_s \in \mathbb{R}^{N_s \times N_s}$, with a Gaussian kernel:

$$\mathbf{K}_s(i,j) = \exp\left(-\frac{\|x_i^s - x_j^s\|_2^2}{\epsilon_s}\right), \tag{2}$$

where $\epsilon_s > 0$ is a scale hyperparameter. By employing this local kernel, which quantifies pairwise local similarities between data samples, the graph effectively captures the local neighborhood relationships. Subsequently, we obtain the source diffusion operator, denoted by $\mathbf{P}_s \in \mathbb{R}^{N_s \times N_s}$, by normalizing the affinity matrix $\mathbf{K}_s$ to be row-stochastic, i.e., ensuring that the sum of each row equals 1. Following this standard normalization (Coifman & Lafon, 2006), the rows of $\mathbf{P}_s$ represent the transition probabilities of a diffusion process on $\mathcal{M}$ defined at the source samples $\{x_i^s\}_{i=1}^{N_s}$.

In the second diffusion step, we apply a cross-domain diffusion, which captures the inter-domain relationships by incorporating both the source and target samples. To this end, we define a bipartite graph $G_c$ with a vertex set $V_c = \{v_1^s, \ldots, v_{N_s}^s, v_1^t, \ldots, v_{N_t}^t\}$ that represents the $N_s$ source samples and $N_t$ target samples. In this graph, each node $v_i^s$ has edges only to target nodes, and each node $v_j^t$ has edges only to source nodes. Consequently, the graph adjacency matrix consists of two off-diagonal blocks. For our purpose, we utilize only the $N_s \times N_t$ top-right block of the matrix. The weight of the edge between source node $v_i^s$ and target node $v_j^t$ is given by:

$$\mathbf{K}_c(i,j) = \exp\left(-\frac{\|x_i^s - x_j^t\|_2^2}{\epsilon_c}\right), \tag{3}$$

where $\epsilon_c > 0$ is a scale hyperparameter. We normalize this kernel matrix to be row-stochastic, and denote the resulting cross-domain diffusion operator by $\mathbf{Q} \in \mathbb{R}^{N_s \times N_t}$.

In the third and last diffusion step, designed to capture intra-domain similarities within the target domain, we construct the graph $G_t$ exclusively from target samples. This graph comprises $N_t$ nodes, representing the target samples, and an affinity matrix $\mathbf{K}_t \in \mathbb{R}^{N_t \times N_t}$, constructed using a Gaussian kernel with scale $\epsilon_t$ similarly to Equation 2. After normalizing $\mathbf{K}_t$ to be row-stochastic, we obtain the target diffusion operator, denoted by $\mathbf{P}_t \in \mathbb{R}^{N_t \times N_t}$.

The final operator $\mathbf{S}$, which serves as the basis for the transport cost, is defined by:

$$\mathbf{S} = \mathbf{P}_s \mathbf{Q} \mathbf{P}_t. \tag{4}$$

By construction, $\mathbf{S}(i, j)$ is the probability of transporting from source sample $x_i^s$ to target sample $x_j^t$ through local neighborhoods. Specifically, $\mathbf{S}(i, j)$ encapsulates the probability of (i) diffusing from source sample $x_i^s$ to its source neighbors, (ii) diffusing from these source neighbors to target samples that are neighbors of $x_j^t$, and finally (iii) diffusing from these target neighbors to $x_j^t$.

Finally, we apply a negative element-wise logarithm to the diffusion probabilities to translate the notion of similarity into a notion of cost, aligning with the requirements of the OT problem. Thus, the proposed transportation cost matrix, referred to as Diffusion-OT, is defined by: $\mathbf{C} = -\log(\mathbf{S})$.

The computational complexity of Diffusion-OT is $\mathcal{O}(n^3)$; see Appendix C.2 for details.

### 4.1 THEORETICAL RESULTS

While competing methods typically do not include theoretical insights, our approach is driven by the following theoretical basis. Specifically, we prove that the proposed operator $\mathbf{S}$ is a diffusion operator and analyze its asymptotic behavior, showing how it aligns with the desired properties. This analysis provides a deeper understanding of the motivation behind our method. Building on this, we now transition to examining the diffusion operator in a continuous setting.

Consider infinite source and target datasets, sampled i.i.d from the probability distributions $\mu$ and $\nu$, respectively, defined over the continuous manifold $\mathcal{M}$. Consider a Gaussian kernel $k_\epsilon(x, y)$ with a scale parameter $\epsilon$. In analogy to the row-stochastic normalization in the discrete case, we define the continuous source diffusion operator $P_{s,\epsilon}$ by:

$$P_{s,\epsilon} f(y) = \int p_{s,\epsilon}(x, y) f(x) \mu(x) dx = \int \frac{k_\epsilon(x, y)}{d_{s,\epsilon}(x)} f(x) \mu(x) dx, \tag{5}$$

where $p_{s,\epsilon}(x, y)$ is the normalized kernel, and $d_{s,\epsilon}(x) = \int k_\epsilon(x, y) \mu(y) dy$.

Similarly, we define the continuous target diffusion operator $P_{t,\epsilon}$, which operates on the target domain, by employing Equation 5 with respect to $\nu$ and with the normalized kernel $p_{t,\epsilon}(x, y) = \frac{k_\epsilon(x,y)}{d_{t,\epsilon}(x)}$, where $d_{t,\epsilon}(x) = \int k_\epsilon(x, y) \nu(y) dy$.

The cross-domain diffusion operator is defined by: $Q_\epsilon f(y) = \int \frac{k_\epsilon(x,y)}{d_{t,\epsilon}(x)} f(x) \mu(x) dx$. Note that in this analysis, without loss of generality, we use the same kernel $k_\epsilon(x, y)$ for all three operators.

In the discrete case, applying the diffusion operator defined in Equation 4 to the source samples necessitates using the transpose of $\mathbf{S}$. Thus, in the continuous case, the diffusion operator $S_\epsilon$, defined as the composition of the three operators $P_{s,\epsilon}$, $Q_\epsilon$, and $P_{t,\epsilon}$, can be expressed as $S_\epsilon f(x) = P_{t,\epsilon} Q_\epsilon P_{s,\epsilon} f(x)$.

**Proposition 1.** Suppose $f \in \mathcal{C}^4(\mathcal{M})$, and suppose $\mu, \nu \in \mathcal{C}^4(\mathcal{M})$ denote the probability measures of the source and target domains, respectively, where $\mu$ is dominated by $\nu$. Denote the Radon–Nikodym (RN) derivative by $\frac{\mu}{\nu}$. For sufficiently small $\epsilon$, the asymptotic expansion of operator $S_\epsilon$ is given by:

$$S_\epsilon f(x) = \frac{\mu}{\nu}(x) \left[ f - \frac{m_2}{m_0} \epsilon \left[ 3\Delta f + 2 \left( f \frac{\Delta\left(\frac{\mu}{\nu}\right)}{\frac{\mu}{\nu}} + 2\nabla f \nabla \log\left(\frac{\mu}{\nu}\right) \right) \right. \right. \tag{6}$$

$$\left. \left. - f\left(\frac{\Delta\mu}{\mu} + 2\frac{\Delta\nu}{\nu}\right) \right] \right] (x) + O(\epsilon^2),$$

where $\Delta, \nabla$ are the Laplace–Beltrami operator and the covariant derivative on $\mathcal{M}$, respectively, and $m_0, m_2$ are two constants defined by the Gaussian kernel and by the manifold $\mathcal{M}$.

The proof appears in Appendix E.

The expression derived in Proposition 1 suggests that the proposed Diffusion-OT method inherently considers both the domain shift and the intra-domain relationships of the source and target domains. Specifically, the proposition implies that the proposed diffusion operator leads to an anisotropic diffusion process on the manifold $\mathcal{M}$, influenced by the manifold geometry (conveyed by the differential operators $\Delta$ and $\nabla$ w.r.t. the manifold), the source and target domains, and the discrepancy between the domains, represented by the RN derivative. In the expression, we can identify the distinct contribution of each term to the diffusion process. The term within the first set of round parentheses, involving both the first and second derivatives of the RN derivative, characterizes the inter-domain relationships. Meanwhile, the term within the second set of round parentheses, incorporating separate expressions of the second derivative of the source and target domains, represents the intra-domain similarities within each domain, as well as the local geometry of the domains. To illustrate this point, we examine a 2D toy example outlined in Appendix C.1.

We remark that if $\mu(x) = \nu(x)$ for all $x \in \mathcal{M}$, indicating no domain shift, the asymptotic expansion of $S_\epsilon$ is given by:

$$S_\epsilon f(x) = f(x) - \frac{m_2}{m_0}\epsilon \left( 3\Delta f - 3f\frac{\Delta\mu}{\mu} \right)(x) + O(\epsilon^2), \tag{7}$$

coinciding with the asymptotic expansion of a diffusion operator on $\mathcal{M}$ with distribution $\mu$ (Coifman & Lafon, 2006), applied three times.

### 4.2 Practical Remarks

To incorporate further adaptability to our cost, we integrate explicit neighborhood information into the intra-domain diffusion operators. The intra-domain kernel is thus defined as follows:

$$\mathbf{K}(i,k) = \begin{cases} \exp\left(-\frac{\|x_i - x_k\|_2^2}{\epsilon}\right) & \text{if } k \in \mathcal{N}_i \\ 0 & \text{otherwise} \end{cases}, \tag{8}$$

where $\mathcal{N}_i$ is defined as the set of all indices $\{k\}$ such that the sample $x_k$ is within the neighborhood of sample $x_i$.

In the context of UDA, we assume access to labeled source samples $\{(x_i^s, y_i^s)\}_{i=1}^{N_s}$. To improve performance, we incorporate source label information into the cost by allowing each source sample $x_i^s$ to diffuse only to other source samples within the same class. Specifically, for each $x_i^s$, the neighborhood $\mathcal{N}_i^s$ is defined to include only source samples with the label $y_i^s$. Accordingly, the label-enhanced source kernel is defined by Equation 8, where $\mathcal{N}_i = \{k \mid y_k^s = y_i^s\}$. This approach stands in contrast to other methods that propose incorporating source labels by adding a regularization term to the problem formulation. Such regularization terms often impose constraints on the OT solver or the learned geometry, potentially restricting the method's adaptability and performance.

For the target operator $\mathbf{P}_t$, or in scenarios where labels are unavailable in a specific DA setup, we define the neighborhood $\mathcal{N}_i$ as the three nearest neighbors of $x_i$. This allows leveraging local structure effectively, even in the absence of label information.

The proposed method is summarized in Algorithm 1, provided in the supplementary materials.

## 5 Experiments

In this section, we provide a visual comparison of Diffusion-OT with competing methods using a toy example (Section 5.1). Additionally, we present experiments on deep domain adaptation tasks (Section 5.2) and non-Euclidean dataset adaptation (Section 5.3), comparing the performance of Diffusion-OT against the respective SOTA methods. The source code is available here[1].

---

[1]The code will be made available on Github upon acceptance.

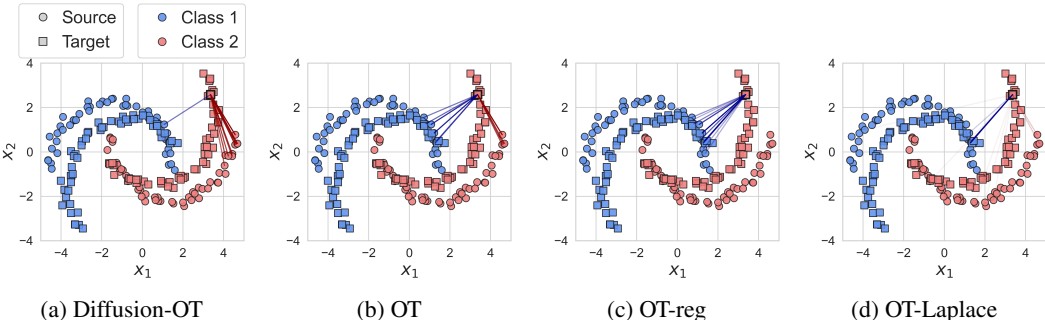

(a) Diffusion-OT        (b) OT        (c) OT-reg        (d) OT-Laplace

Figure 1: Two moons illustration for a $40°$ rotation angle. Source samples are marked with circles, target samples with squares, and the colors of the markers indicate the labels. Each line represents an entry $\gamma(i, j)$ in the transportation plan, where the line color corresponds to the label of the source sample $x_i^s$, and the line intensity reflects the values in the plan. (Best viewed in colors).

### 5.1 TWO MOONS: VISUAL ILLUSTRATION

We utilize the toy example presented in Courty et al. (2016), which involves two entangled moons as the source domain, each representing a different class, where the target domain is created by rotating the source data. As the rotation angle increases, the adaptation problem becomes more challenging. For additional details on the data, refer to Appendix D.1.

We employ Diffusion-OT with the entropy-regularized OT problem (Cuturi, 2013), as presented in Equation 1, to derive the transport plan. For comparison, we present results using the baseline method, which employs the entropy-regularized OT problem solved using the standard cost, referred to simply as OT. Additionally, due to the limited number of methods proposing alternative transport costs – particularly those that integrate source label information and can be applied in non-deep frameworks – we include two regularization approaches: OT-reg (Courty et al., 2016; 2014), which incorporates source labels, and OT-Laplace (Courty et al., 2016; Flamary et al., 2014), which uses a Laplace regularization term to preserve data structure. We emphasize that all three methods use the squared Euclidean distance as the transportation cost. Table 4 in Appendix D.1 summarizes the key differences between Diffusion-OT and these three competing methods.

In Figure 1, we visualize the plans obtained using the aforementioned methods for a $40°$ rotation angle, focusing on one target sample at the top edge of the red moon, marked by a bold black square. For clarity, we plot lines from this target sample to the ten source samples with the highest values in the corresponding column of the plan. In Figure 1a, we observe that Diffusion-OT has achieved a nearly optimal plan, with almost all source samples contributing mass to the red-labeled target sample belonging to the red class, except for one. In OT (Figure 1b), since the transportation cost is based solely on the pairwise Euclidean distances, the plan results in undesirable mapping, where most of the mass contributed to the chosen sample comes from source samples associated with the blue class, even though the target sample is red-labeled. In OT-reg (Figure 1c), the regularization term promotes group sparsity, encouraging a solution where each target sample receives mass only from source samples within the same class. In this case, it results in the most undesirable plan, where all the source samples contributing mass to the chosen target sample are from the opposite class. In OT-Laplace (Figure 1d), the core idea is that samples with small Euclidean distances before transportation should remain close after transportation. We observe that the plan merely mirrors the behavior of the standard OT. The neighborhood information encoded in the regularization is insufficient to correct the Euclidean cost. We note that there are fewer than ten lines because this method does not include entropy regularization, resulting in a sparser transportation plan.

This scenario highlights Diffusion-OT's effectiveness in handling challenging mappings. Diffusion-OT leverages source labels not as strict constraints, as OT-reg, but to guide the diffusion process in the source domain, while still accounting for the local geometry of the data. Additionally, it shows that by taking into account intra-domain relationships, rather than relying solely on pairwise Euclidean distances, the chosen sample can receive mass from source samples that are remote in Euclidean terms, as the diffusion distance is effectively incorporated. Indeed, we see that Diffusion-

OT successfully connects the chosen target sample to the correct, more distant source sample, which no other method considered. We provide further analysis in Appendix D.1, including a second scenario (Figure 3), an illustrative examination of the relationship between the diffusion process and the resulting plan (Figure 4), and a comparison of the target classification error rates across different angles (Table 5).

## 5.2 DEEP DOMAIN ADAPTATION

In the following experiments, we incorporate Diffusion-OT into the DeepJDOT framework, proposed by Damodaran et al. (2018). We apply our method to three benchmark domain adaptation datasets: Digits, Office-Home, and VisDA. One of the key advantages of our method is its adaptability to any OT formulation. To demonstrate this flexibility, we employ different formulations for each dataset, as detailed below, resulting in a different baseline method across datasets. For additional information regarding the datasets, framework, training process, model architecture, and parameters, please refer to Appendix D.2.

**Baselines and competing methods.** In addition to DeepJDOT (Damodaran et al., 2018), which utilizes the framework with the standard transport cost, we present comparisons to more recent methods that leverage the DeepJDOT framework. However, unlike DeepJDOT, which employs the network simplex flow algorithm, these approaches utilize different OT problem formulations: Unbalanced OT (UOT), referred to as JUMBOT (Fatras et al., 2021), and Partial OT (POT), termed as m-POT (Nguyen et al., 2022a), to derive the optimal plan. Notably, all three methods use the squared Euclidean distance as the transportation cost and differ solely in their approach to obtaining $\gamma$. We also compare our approach to two other OT-based deep models, which propose to employ a different transportation cost function from the conventional squared Euclidean distance, namely ETD (Li et al., 2020) and RWOT (Xu et al., 2020). In addition to OT-based methods, we benchmark our method against established DA baselines, including DANN (Ganin et al., 2016) and ALDA (Chen et al., 2020). For DeepJDOT (Damodaran et al., 2018), JUMBOT (Fatras et al., 2021), and m-POT (Nguyen et al., 2022a), all employing the DeepJDOT framework with different OT problem formulations, we reproduce the results ourselves using the POT (Python Optimal Transport) package (Flamary et al., 2021), an open-source Python library. Results for DANN (Ganin et al., 2016) and ALDA (Chen et al., 2020) are sourced from Nguyen et al. (2022a). For the remaining methods, we rely on the reported results from their respective papers.

**Results.** Table 1a presents the results for the Digits datasets. Details regarding the model architecture and parameters for each method are available in Appendix D.2.1. For the digits dataset, we follow Fatras et al. (2021) and use the UOT formulation to obtain the transportation plan. In this case, JUMBOT serves as the baseline, and we refer to our method using this formulation as Diffusion-UOT. The table displays the mean and standard deviation of the target test set over three runs. Results for each individual run can be found in Table 6 in Appendix D.2.1. The highest accuracy is indicated in bold, while the second highest is underlined. Our method achieves the highest accuracy for SVHN→MNIST and USPS→MNIST. Additionally, it attains the second highest accuracy for MNIST→USPS, followed by RWOT. We acknowledge that this dataset is considered an easy DA task, and it appears that the generator learns features that are well separated by the classes, even when using the standard cost. Therefore, the obtained minor improvement is expected.

Table 2 shows the results for the Office-Home dataset. In this experiment, we again incorporate the proposed cost into the UOT problem formulation to derive the optimal plan, referring to our method as Diffusion-UOT. Here, DeepJDOT and JUMBOT were reproduced using the original DeepJDOT framework, while TS-POT was reproduced using a two-stage (TS) implementation, as proposed in the original paper (Nguyen et al., 2022a). The results are averaged over three runs, with full details, including the parameters used for each method, provided in Appendix D.2.2. Table 2 summarizes the target test accuracy for each scenario of the Office-Home dataset. Remarkably, the proposed method achieves the highest average accuracy across the 12 scenarios and outperforms the tested existing methods in 10 out of the 12 scenarios. Specifically, it surpasses the baseline JUMBOT, which was implemented using the UOT formulation with the standard cost, by more than 2%.

Table 1b presents the mean and standard deviation over three runs for the VisDA dataset. In this experiment, we integrate Diffusion-OT into the POT problem formulation, using the TS implementation proposed by Nguyen et al. (2022a). Consequently, TS-POT serves as the baseline, and we refer

Table 1: Target domain accuracy. (a) Results on the Digits dataset, where "Ours" denotes Diffusion-UOT. (b) Results on the VisDA dataset, where "Ours" denotes TS-Diffusion-POT.

(a)

| Method | SVHN-MNIST | USPS-MNIST | MNIST-USPS | Avg |
|---|---|---|---|---|
| DANN | $95.80 \pm 0.29$ | $94.71 \pm 0.12$ | $91.63 \pm 0.53$ | 94.05 |
| ALDA | $98.81 \pm 0.08$ | $98.29 \pm 0.07$ | $95.29 \pm 0.16$ | 97.46 |
| ETD | $97.9 \pm 0.4$ | $96.3 \pm 0.1$ | $96.4 \pm 0.3$ | 96.9 |
| RWOT | $98.8 \pm 0.1$ | $97.5 \pm 0.2$ | $\mathbf{98.5 \pm 0.2}$ | 98.3 |
| DeepJDOT | $96.04 \pm 0.66$ | $97.22 \pm 0.23$ | $86.12 \pm 0.60$ | 93.13 |
| JUMBOT | $98.99 \pm 0.06$ | $98.68 \pm 0.08$ | $96.93 \pm 0.42$ | 98.20 |
| m-POT | $98.98 \pm 0.08$ | $98.63 \pm 0.13$ | $96.04 \pm 0.02$ | 97.88 |
| Ours | $\mathbf{99.19 \pm 0.04}$ | $\mathbf{98.87 \pm 0.03}$ | $97.81 \pm 0.30$ | $\mathbf{98.62}$ |

(b)

| Method | Accuracy |
|---|---|
| DANN | $67.63 \pm 0.34$ |
| ALDA | $71.22 \pm 0.12$ |
| DeepJDOT | $69.58 \pm 0.34$ |
| JUMBOT | $72.97 \pm 0.26$ |
| TS-POT | $75.65 \pm 0.78$ |
| Ours | $\mathbf{78.56 \pm 0.15}$ |

Table 2: Target domain accuracy on the Office-Home dataset.

| Method | A-C | A-P | A-R | C-A | C-P | C-R | P-A | P-C | P-R | R-A | R-C | R-P | Avg |
|---|---|---|---|---|---|---|---|---|---|---|---|---|---|
| ResNet-50 | 34.9 | 50.0 | 58.0 | 37.4 | 41.9 | 46.2 | 38.5 | 31.2 | 60.4 | 53.9 | 41.2 | 59.9 | 46.13 |
| DANN | 47.9 | 67.1 | 74.9 | 53.8 | 63.5 | 66.4 | 53.0 | 44.4 | 74.4 | 65.5 | 53.0 | 79.4 | 61.93 |
| ALDA | 54.0 | 74.9 | 77.1 | 61.4 | 70.6 | 72.8 | 60.3 | 51.0 | 76.7 | 67.9 | 55.9 | 81.9 | 67.04 |
| ETD | 51.3 | 71.9 | $\mathbf{85.7}$ | 57.6 | 69.2 | 73.7 | 57.8 | 51.2 | 79.3 | 70.2 | 57.5 | 82.1 | 67.29 |
| RWOT | 55.2 | 72.5 | 78.0 | 63.5 | 72.5 | 75.1 | 60.2 | 48.5 | 78.9 | 69.8 | 54.8 | 82.5 | 67.63 |
| DeepJDOT | 52.0 | 70.9 | 76.1 | 60.5 | 66.6 | 69.2 | 58.4 | 48.7 | 75.3 | 68.9 | 54.9 | 79.9 | 65.12 |
| JUMBOT | 55.7 | 75.0 | 80.7 | 65.1 | 74.5 | 75.1 | 65.3 | 53.3 | 79.6 | 74.5 | 59.3 | 83.9 | 70.17 |
| TS-POT | $\mathbf{57.4}$ | 77.1 | 81.6 | 68.3 | 72.8 | 76.5 | 67.4 | 55.1 | 80.6 | 75.4 | 59.9 | 84.0 | 71.36 |
| Diffusion-UOT | 57.2 | $\mathbf{78.0}$ | 82.1 | $\mathbf{70.2}$ | $\mathbf{74.9}$ | $\mathbf{78.8}$ | $\mathbf{68.1}$ | $\mathbf{56.5}$ | $\mathbf{82.0}$ | $\mathbf{75.6}$ | $\mathbf{60.9}$ | $\mathbf{84.8}$ | $\mathbf{72.43}$ |

to our method as TS-Diffusion-POT. DeepJDOT and JUMBOT were reproduced using the original DeepJDOT framework, while TS-POT was reproduced using the TS implementation. Additional details regarding the TS implementation and the parameters used for each method are available in Appendix D.2.3. We observe that employing the TS implementation with the POT formulation is highly effective for this dataset, as TS-POT obtained significantly higher accuracy than the other competing methods. Notably, incorporating Diffusion-OT into this framework further enhances performance, increasing accuracy by nearly 3% on average. In Figure 5 in Appendix D.2.3, we analyze the t-SNE representation of the VisDA target features obtained from the deep model trained with Diffusion-OT, comparing them to those learned using the baseline method, TS-POT.

## 5.3 DOMAIN ADAPTATION FOR NON-EUCLIDEAN DATA

In this section, we depart from employing OT to learn domain-invariant features and instead directly utilize the optimal plan to transport source features into the target domain. We evaluate the proposed method on electroencephalogram (EEG) data, addressing both binary and multi-class classification tasks. Specifically, we apply our approach to two Motor Imagery (MI) benchmarks from the BCI Competition IV. The first dataset, dataset I from Blankertz et al. (2007), is referred to as MI1, and the second dataset, IIa from Tangermann et al. (2012), is referred to as MI2. We follow the methodologies of Barachant et al. (2011; 2013); Zanini et al. (2017); Rodrigues et al. (2018); Yair et al. (2019), and solve the DA problem on the (non-Euclidean) Riemannian manifold of Symmetric Positive Definite (SPD) matrices. For all experiments in this section, we utilize the entropy-regularized problem formulation (Cuturi, 2013), referring to our methods as Diffusion-OT. The baseline method, which uses this formulation with the standard transport cost, is labeled as OT in the tables. Additionally, we include a comparison with OT-reg (Courty et al., 2016; 2014), a regularization approach that incorporates source labels into the problem formulation, which we consider as a secondary baseline. For detailed information on the framework, datasets, pre-processing, and parameters used, please refer to Appendix D.3.

Table 3: Binary classification accuracy on two BCI datasets. (a) Results for the cross-subject task. (b) Results for the leave-one-subject-out task.

(a)

| Method | MI1 | MI2 | Avg |
|---|---|---|---|
| CSP-LDA | 57.23 | 58.70 | 57.97 |
| RA-MDRM | 64.98 | 66.60 | 65.79 |
| EA-CSP-LDA | 66.85 | 65.00 | 65.93 |
| MEKT-R | 70.99 | 68.73 | 69.86 |
| METL | _71.81_ | _69.06_ | _70.44_ |
| OT | 63.83 | 64.59 | 64.21 |
| OT-reg | 67.17 | 67.26 | 67.22 |
| Diffusion-OT | **72.24** | **70.41** | **71.33** |

(b)

| Method | MI1 | MI2 | Avg |
|---|---|---|---|
| CSP-LDA | 59.71 | 67.75 | 63.73 |
| RA-MDRM | 69.21 | 70.91 | 70.06 |
| EA-CSP-LDA | 79.79 | 73.53 | 76.66 |
| MEKT-R | 83.42 | _76.31_ | _79.87_ |
| METL | 83.14 | 76.00 | 79.57 |
| WBT | _85.21_ | 74.31 | 79.76 |
| Diffusion-WBT | **87.29** | **77.24** | **82.26** |

**Results.** To facilitate a meaningful comparison with SOTA methods limited to two-class scenarios, we evaluate our proposed method on a binary classification task. In Appendix D.3.2, we present multi-class classification experiments compared against baseline methods. For this evaluation, we utilize both BCI datasets; specifically, for the MI2 dataset, we select the left hand (class 1) and the right hand (class 2) as the two classes of interest. Our method is benchmarked against baseline algorithms commonly employed in BCI classification, namely CSP-LDA (Grosse-Wentrup & Buss, 2008), RA-MDRM (Zanini et al., 2017), and EA-CSP-LDA (He & Wu, 2019). Additionally, we compare our approach to two SOTA methods: MEKT (Zhang & Wu, 2020) and METL (Cai et al., 2022). Results for CSP-LDA, RA-MDRM, EA-CSP-LDA, and METL are obtained from the tables presented in Cai et al. (2022), while results for MEKT are sourced from the original paper (Zhang & Wu, 2020).

Table 3a presents the results for the cross-subject task. In this experiment, each subject, in turn, serves as the target domain, with the remaining subjects alternately acting as the source domain. Accuracy is computed by training a linear SVM classifier on the transformed source mappings and evaluating it on the target mappings. The results show the average target accuracy across source subjects, averaged over all subjects. The proposed method achieved the highest accuracy for both datasets, improving the baselines OT and OT-reg by 7% and 4%, respectively. Detailed subject-wise results appear in Table 9 and Table 10 in Appendix D.3.1, along with the experimental parameters.

In Table 3b, we present the results for the leave-one-subject-out task, where each subject alternately acts as the target domain while all others are used as source domains. We apply the multi-source OT-based DA framework from Montesuma & Mboula (2021), referred to as WBT, and denote our method as Diffusion-WBT. The table reports the average target accuracy, computed by training a linear SVM classifier on all the transported source representations and evaluating it on the target samples. Interestingly, the baseline method (WBT with the standard cost) outperformed all competing methods on dataset MI1. Notably, incorporating the proposed cost resulted in further accuracy improvements, achieving the highest performance across both datasets. For details on the framework and the experimental parameters, see Appendix D.3.1.

## 6    CONCLUSIONS

In this work, we introduce a novel transportation cost of OT for DA, constructed using diffusion geometry tools. Our experimental results highlight that Diffusion-OT achieves superior performance compared to other methods across various UDA tasks. We acknowledge a fundamental limitation of our method; since it incorporates inter-domain relationships, it operates under the assumption that both source and target domains reside in the same space. While this assumption is common to all DA methods, we believe that Diffusion-OT has the potential to be extended to other applications, which we view as directions for future work.

## 7 ETHICS STATEMENT

We adhere to the ICLR Code of Ethics and ensure that our research practices maintain the highest standards of integrity. The datasets utilized in our experiments are publicly available, completely anonymized, and do not include any personal information. We do not foresee any negative societal impact arising from our research. Furthermore, we ensure transparency in our methodologies and report our findings honestly. We have no conflicts of interest to disclose related to this research.

## 8 REPRODUCIBILITY STATEMENT

The source code will be made available on GitHub upon acceptance. To ensure reproducibility, the datasets used in our experiments, along with detailed preprocessing steps, are outlined in Appendix D, which also includes information on the architecture, hyperparameters, training protocols, and more. Our theoretical foundations, including the assumptions and proofs regarding the proposed operator $\mathbf{S}$, can be found in Appendix E.

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

# SUPPLEMENTARY MATERIALS

## A RELATED WORK

In the OTDA framework, proposed by Courty et al. (2016), the authors suggest the integration of several regularization terms to improve performance, including a class-based regularization term (Courty et al., 2014) that utilizes source label information. By employing specific parameter configurations, the optimization problem can be effectively addressed using the Sinkhorn algorithm (Cuturi, 2013). The authors also proposed applying graph regularization on the transported samples (Flamary et al., 2014). While this regularization, termed Laplace regularization, considers neighborhood information prior to transportation, it is limited to Euclidean geometry. Furthermore, their optimization problem is quadratic, rendering it unsolvable by efficient algorithms and resulting in high computational complexity. Consequently, it is impractical for incorporation into deep models or application to large datasets.

DeepJDOT (Damodaran et al., 2018) enhances the OTDA framework by employing deep models. This approach addresses two significant drawbacks in the OTDA framework. Firstly, handling large datasets becomes infeasible with OTDA due to the quadratic increase in complexity with the number of samples $n$, and the fact that it involves computations with $n \times n$ matrices. Conversely, DeepJDOT can train deep models using mini-batches, making it suitable for large datasets. Secondly, OTDA predominantly employs the squared Euclidean distance as the transportation cost, and, as a result, it often performs poorly on non-Euclidean datasets. In contrast, although the DeepJDOT framework also uses the squared Euclidean distance as the cost, it operates in a latent space learned by a deep model, potentially enabling superior performance on non-Euclidean data.

Few papers in the literature have proposed alternative cost functions diverging from the standard distance metric. For instance, Tai et al. (2021) introduced an innovative approach termed SLA, which utilizes the OT formulation as a novel label assignment method. In this approach, each element $(i, k)$ in the transportation cost matrix is associated with the probability that sample $i$ belongs to class $k$. The method presented in Gu et al. (2022) is a semi-supervised approach that assumes the availability of a set of source-target pairs, termed keypoints. Using these keypoints, the authors propose to compute a relation score for all source and target samples, which is subsequently used to construct the cost matrix. A related semi-supervised approach is presented in Duque et al. (2023) for manifold alignment. This method similarly relies on prior correspondence knowledge between some source and target samples to align the domains, with inter-domain distances computed solely from these corresponding pairs. While they propose applying a diffusion process to capture intra-domain relationships, they later compute the Cosine distance between submatrices of the obtained probability matrices, which serves as their final cost. Consequently, their semi-supervised transportation cost lacks a notion of probability. The Gromov-Wasserstein distance is typically used to align distributions by leveraging only their intra-domain pairwise similarities. However, some works have proposed innovative approaches to incorporate inter-domain relationships as well. For example, Yan et al. (2018) introduce a semi-supervised DA method that adds a regularization term to the optimization problem. This term incorporates cross-domain distances, computed in the target domain after transportation, and relies on target labels. Alternatively, Xu et al. (2019) present a graph-matching method that learns an embedding space for the nodes, enabling the computation of inter-domain relationships, which are included in the optimization via an additional term.

In the context of UDA, Li et al. (2020) introduced ETD, which utilizes a weighted Euclidean distance for the cost function. They proposed leveraging an attention mechanism to learn the correlation between source and target samples, subsequently using this correlation as the weights in their proposed transportation cost. The OT distance is then employed as the training objective, serving as a quantification of the domain discrepancy. The work in Xu et al. (2020) also proposed using a weighted Euclidean distance as the transportation cost. In that method, termed RWOT, the element $(i, k)$ in the weight matrix quantifies the probability that sample $i$ belongs to class $k$. This probability is dynamically computed based on both spatial prototypical information and the pseudo-classification probability of target samples.

## B BACKGROUND ON DIFFUSION GEOMETRY

Let $\{x_i\}_{i=1}^N$ be a given set of data samples on a hidden manifold $\mathcal{M}$ with a metric $g$. Constructing a graph is a common strategy for approximating the manifold's structure (Roweis & Saul, 2000; Tenenbaum et al., 2000; Belkin & Niyogi, 2003; Coifman & Lafon, 2006). By leveraging a local kernel, which quantifies pairwise local similarities between data samples, this graph captures local neighborhood relationships, facilitating effective representation of the underlying manifold geometry. Consider a weighted graph $G = (V, E)$ with a vertex set $V = \{v_1, \ldots, v_N\}$, where each node $v_i \in V$ corresponds to the sample $x_i$. Let $\mathbf{K} \in \mathbb{R}^{N \times N}$ be an affinity matrix, whose $(i, j)$-th element encodes the weight of the edge between nodes $v_i$ and $v_j$ and is given by:

$$\mathbf{K}(i, j) = h\left(\frac{d_g^2(x_i, x_j)}{\epsilon}\right),\tag{9}$$

where $d_g(\cdot, \cdot)$ is a distance function induced by the metric $g$, $\epsilon > 0$ is a scale hyperparameter, and $h$ is a positive function with exponential decay.

Next, the affinity matrix $\mathbf{K}$ is normalized to be row-stochastic:

$$\mathbf{P}(i, j) = \frac{\mathbf{K}(i, j)}{d(i)},\tag{10}$$

where $d(i) = \sum_j \mathbf{K}(i, j)$. The matrix $\mathbf{P}$ is often viewed as a diffusion operator (Coifman & Lafon, 2006), as $\mathbf{P}(i, j)$ defines a transition probability from node $i$ to node $j$ of a Markov chain on the graph. Analogously, $\mathbf{P}$ can be viewed as the transition probabilities of a diffusion process defined at the data samples $\{x_i\}_{i=1}^N$ on the manifold $\mathcal{M}$ (Coifman & Lafon, 2006).

## C DIFFUSION OPTIMAL TRANSPORT – ADDITIONAL INFORMATION

### C.1 THE DIFFUSION PROCESS – SIMULATION

In this section, we illustrate the behavior of the proposed diffusion operator, and more specifically, the expression outlined in Proposition 1, through a toy example. Consider the 2D space $[0, 1]^2$, where the source domain $\mu$ is represented by a uniform distribution over the entire space, and the target domain $\nu$ is concentrated along a 1D contour. Specifically, 20% of the target data are sampled from a uniform distribution over $[0.1]^2$. For the remaining 80% of the target samples, the coordinates are generated as follows. The first dimension $x_1$ is computed as the sum of two variables sampled from uniform distributions: $u_1 \in U[0.02, 0.98]$ and $u_2 \in U[-0.02, 0.02]$, giving $x_1 = u_1 + u_2$. The second dimension $x_2$ is generated with $x_2 = 0.4\sin(2\pi u_1) + 0.5$, where $u_1$ is the same variable used for $x_1$. Figure 2a presents the target domain, estimated using Kernel Density Estimation (KDE).

To illustrate the behavior of the proposed operator, we compute the diffusion operator $\mathbf{S}$, which contains the probabilities of transporting from the uniform source samples to the target samples concentrated on the contour, as presented in Equation 4. Figure 2b visualizes the resulting vector field on the space $[0, 1]^2$, as induced by $\mathbf{S}$. As expected, we observe that for all source samples, the average direction induced by the diffusion operator is toward the 1D contour. In Figures 2c and 2d, we examine two specific cases. Figure 2c demonstrates the proposed diffusion process initiated at a source sample located on the contour with high target probability density, marked by the black star in the figure. As the source distribution is uniform, by definition, the RN derivative value at this initial sample is small. We observe an anisotropic diffusion that rapidly spreads along the contour where the target density is high. In Figure 2d, we illustrate the proposed diffusion process, with initialization at a source sample located off the contour. Similar to the previous case, we observe an anisotropic diffusion directed toward regions with high probability density. In this scenario, due to the location of the starting point, we see diffusion propagation towards the contour, resulting in a noticeable asymmetry in the diffusion process.

This example demonstrates that the anisotropic diffusion implicitly balances inter-domain and intra-domain relationships. For instance, in Figure 2c, when the chosen source sample is located in a dense region of the target domain – indicating a high probability of belonging to the target – the diffusion

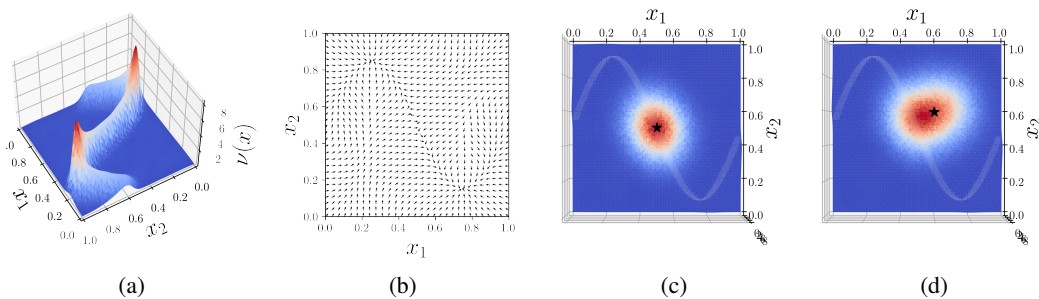

(a)  (b)  (c)  (d)

Figure 2: (a) The PDF of the target domain, colored by PDF values. (b) Visualization of the vector field induced by the proposed diffusion operator. (c) and (d) show the PDF of the target domain projected to 2D, colored by the proposed diffusion process initiated at high-density and low-density regions, respectively. (Best viewed in colors).

assigns more weight to the geometry of the target domain, spreading along the contour. In contrast, in Figure 2d, when the source sample is outside the contour where the target distribution is sparse, the anisotopic diffusion first draws it toward the denser region, utilizing inter-domain relations.

### C.2 COMPUTATIONAL COMPLEXITY OF DIFFUSION-OT

The most commonly used transport cost, the pairwise squared Euclidean distances, has a computational complexity of $\mathcal{O}(n^2)$, assuming balanced datasets. In contrast, our proposed cost involves the following steps:

- Computing pairwise distances for three matrices: $\mathcal{O}(n^2)$.

- Applying the exponential function to the matrices: $\mathcal{O}(n^2)$.

- Normalizing the matrices to be row-stochastic: $\mathcal{O}(n^2)$.

- Multiplying the three probability matrices: $\mathcal{O}(n^3)$.

- Applying the logarithm to the final diffusion operator: $\mathcal{O}(n^2)$.

As noted in Appendix D, we often use doubly-stochastic normalization instead of row-stochastic normalization, employing the Sinkhorn algorithm. Since the complexity of Sinkhorn is also $\mathcal{O}(n^2)$, this does not increase the overall complexity. In total, the computational complexity of the proposed cost is $\mathcal{O}(n^3)$. While higher than the traditional cost, our method avoids the need for a regularization term, which can often add to the optimization complexity.

Additionally, we note that up to the fourth step, the computation involves three $n \times n$ matrices rather than one, resulting in higher memory requirements.

---

**Algorithm 1** Diffusion-OT

---

**Input**: $\{((x_i^s, y_i^s)\}_{i=1}^{N_s}, \{x_j^t\}_{j=1}^{N_t}, \epsilon_s, \epsilon_c, \epsilon_t$.

1: Compute the intra-domain kernel of the source, $\mathbf{K}_s$, using Equation 8 and supervised neighborhood information.
2: Compute the intra-domain kernel of the target, $\mathbf{K}_t$, using Equation 8 with a three nearest neighbors neighborhood.
3: Compute the cross-domain kernel $\mathbf{K}_c$ according to equation 3.
4: Compute the row-stochastic probability matrices, $\mathbf{P}_s, \mathbf{P}_t$ and $\mathbf{Q}$, according to Equation 10.
5: Compute the diffusion matrix $\mathbf{S}$ according to Equation 4.
6: Apply element-wise negative logarithm $\mathbf{C} = -\log(\mathbf{S})$.
7: Return the cost matrix $\mathbf{C}$.

---

# D EXPERIMENTAL DETAILS

**General remarks.** For all experiments, we employed the label-enhanced kernel for the source diffusion operator:

$$\mathbf{K}_s(i,k) = \begin{cases} \exp\left(-\frac{\|x_i^s - x_k^s\|_2^2}{\epsilon_s}\right) & \text{if } y_k^s = y_i^s \\ 0 & \text{otherwise} \end{cases}. \tag{11}$$

For all three diffusion operators $\mathbf{P}_s, \mathbf{Q}, \mathbf{P}_t$, before computing the Gaussian kernels, we conducted median normalization on the distance matrix. After computing the kernels, we performed doubly-stochastic normalization on the square matrices $\mathbf{P}_s$ and $\mathbf{P}_t$ using the Sinkhorn algorithm, as suggested in Landa et al. (2021). The matrix $\mathbf{Q}$, which may not necessarily be square, was normalized to be row-stochastic. Unless specified otherwise, we employed the Sinkhorn algorithm (Cuturi, 2013) to solve the OT problem, utilizing the publicly available POT package (Flamary et al., 2021). All the experiments were conducted on Nvidia DGX A100.

## D.1 TWO MOONS: VISUAL ILLUSTRATION

In this experiment, we analyze the resulting transportation plan and evaluate the performance of Diffusion-OT compared to competing methods using the toy example from Courty et al. (2016). We use sklearn to generate two entangled moons, each with 50 samples and Gaussian noise added with a standard deviation of 0.05. The target domain is generated by rotating the source data. As the rotation angle increases, the adaptation problem becomes more challenging.

We compare Diffusion-OT with the entropy-regularized OT problem (Courty et al., 2016; Cuturi, 2013), referred to as OT, which serves as the baseline method. Additionally, due to the limited number of methods proposing alternative transport costs – particularly those that integrate source label information and can be applied in non-deep frameworks – we include two regularization approaches: OT-reg (Courty et al., 2016; 2014), which incorporates source label information, and OT-Laplace (Courty et al., 2016; Flamary et al., 2014), which uses a Laplace regularization term to preserve data structure. Notably, all three methods use the squared Euclidean distance as the transportation cost. Table 4 summarizes the key differences between Diffusion-OT and these three competing methods. While the first two characteristics in the table are straightforward, the last two may require further clarification. OT-Laplace's optimization relies on simplifying the regularization term under the assumption of Euclidean geometry. For non-Euclidean geometries, the intra-domain distances, inter-domain distances, and barycentric mapping become too complex to optimize, limiting it to Euclidean settings. While the squared Euclidean distance is commonly used as the transport cost in OT and OT-reg, both methods can handle non-Euclidean geometries, as demonstrated in Section 5.3. The specific optimization processes proposed for OT-reg and OT-Laplace restrict their applicability to other formulations or solvers. In contrast, our Diffusion-OT, which is a new transport cost rather than a regularization term, is versatile and applicable to any problem formulation or solver, as shown in Section 5.2.

Table 4: Summary of key differences between Diffusion-OT and competing methods.

|  | OT | OT-reg | OT-Laplace | Diffusion-OT |
|---|---|---|---|---|
| Intra-domain relationship integration |  |  | ✓ | ✓ |
| Label information utilization |  | ✓ |  | ✓ |
| Supports non-Euclidean geometry | ✓ | ✓ |  | ✓ |
| Versatile OT compatibility | ✓ |  |  | ✓ |

To analyze the differences between the methods, we visualize the obtained plans for a $40°$ rotation angle, focusing on two arbitrary target samples that demonstrate the typical advantages of Diffusion-OT over the other methods. We obtain the optimal plan for Diffusion-OT and the other three methods as follows. For Diffusion-OT, we first compute the proposed cost and then use the entropy-regularized OT problem (Cuturi, 2013) to derive the transportation plan $\gamma$. For the other methods, we calculate the standard transportation cost – the squared Euclidean pairwise distances – and then

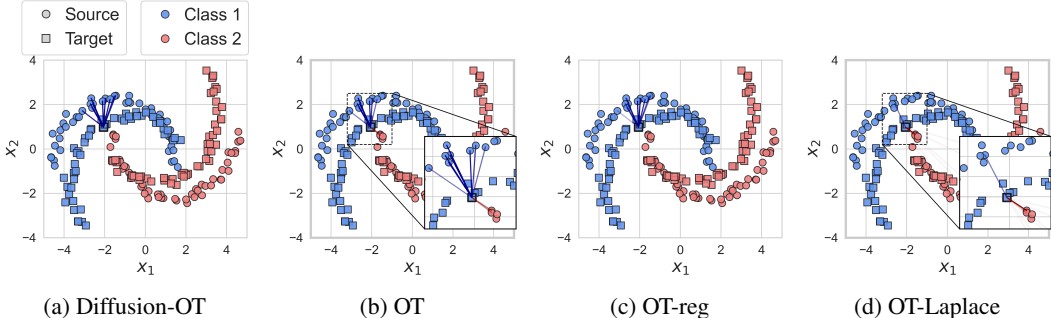

(a) Diffusion-OT      (b) OT      (c) OT-reg      (d) OT-Laplace

Figure 3: Two moons illustration for a $40°$ rotation angle. Source samples are marked with circles, target samples with squares, and the colors of the markers indicate the labels. Each line represents an entry $\gamma(i, j)$ in the transportation plan, where the line color corresponds to the label of the source sample $x_i^s$, and the line intensity reflects the values in the plan. (Best viewed in colors).

solve the OT problem using entropy regularization, label regularization, and Laplace regularization to obtain the transportation plans for OT, OT-reg, and OT-Laplace, respectively.

Figure 3 presents the plan values for a target sample located in the middle of the blue moon, marked by a bold black square. For clarity, we plot lines from this target sample only to ten source samples with the highest values in the corresponding column of the plan. In Diffusion-OT (Figure 3a), we observe the desired optimal plan, where all lines connected to the chosen target sample originate from the same blue class, even though a red-labeled source sample is closer. This outcome is due to incorporating label information into the diffusion process. In OT (Figure 3b), the transportation cost is based solely on the pairwise Euclidean distances between source samples (circles) and target samples (squares). As a result, the blue-labeled target sample receives mass from the closest source samples, including those associated with the other red class. In OT-reg (Figure 3c), the regularization term promotes group sparsity, encouraging a solution where each target sample receives mass only from source samples within the same class. As a result, this method also provides the desired plan in this case. In OT-Laplace (Figure 3d), the core idea is that samples with small Euclidean distances before transportation should remain close after transportation, and vice versa. As a result, the blue-labeled target sample receives mass not only from the nearby red-labeled source sample, but also from a more distant blue-labeled source sample. We note that there are fewer than ten lines because this method does not include entropy regularization, resulting in a sparser transportation plan.

**Cost analysis.** To illustrate the connection between the diffusion process and the resulting transportation plan, we present in Figure 4a the average direction of transport from each target point to the source domain, as induced by the diffusion operator $\mathbf{S}$, for a $60°$ rotation angle. Although the standard OT cost lacks a probability notion, we present a comparable visualization in Figure 4b, by applying the exponential function to the negative cost matrix and normalizing the resulting matrix to be column-stochastic. The diffusion induced by the cost aligns with the resulting plan values depicted in Figures 1 and 3. For example, all target points within the blue dashed ellipse in Figure 4a are directed toward the correct red moon of the source. Conversely, in Figure 4b, most target points are directed toward the closer but incorrect blue moon of the source. Similarly, all target points within the red dashed ellipse in Figure 4a, including the chosen point shown in Figure 3, are directed toward the correct blue moon of the source, even though closer source samples, labeled red, are present. It is worth noting that, while the presented directions correspond to the target samples, the Diffusion-OT cost relies solely on source label information. In contrast, in Figure 4b, the same target points are directed incorrectly, with many directed toward the wrong red source samples. For clarity, only these two areas are highlighted in the figures; however, additional regions, such as the bottom-left edge of the blue target moon and the center of the red target moon, could also be analyzed similarly. This example demonstrates how the proposed method effectively controls the resulting transportation plan by guiding the diffusion process toward the desired directions.

**Classification results.** In addition to the visual illustration, we evaluate the performance of the methods across various rotation angles. Table 5 presents the target classification error rates for these different angles. For the evaluation, we follow the OTDA framework (Courty et al., 2016).

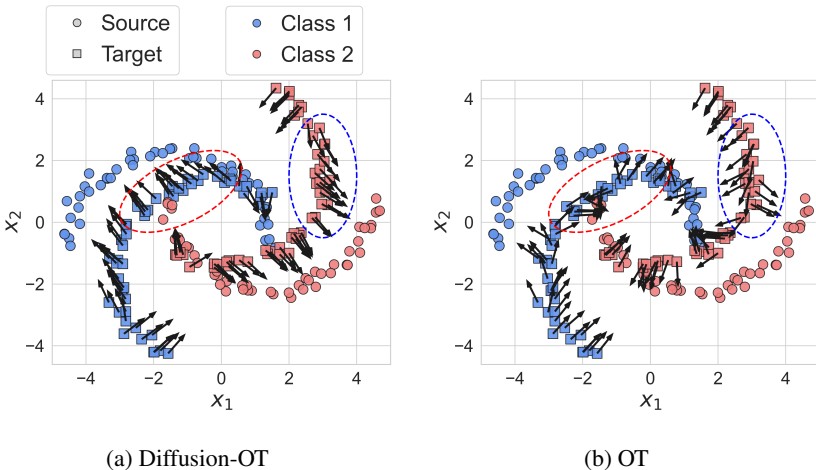

(a) Diffusion-OT  (b) OT

Figure 4: Visualization of the vector field induced by the Diffusion-OT cost and the standard OT cost, for a $60°$ rotation angle. (Best viewed in colors).

This involves computing the optimal plan using Diffusion-OT or one of the competing methods, transporting the source samples to the target domain via barycentric mapping, and then training an SVM classifier with a Gaussian kernel on the source data. The classifier is subsequently tested on the target samples. It is evident that Diffusion-OT achieves significantly higher accuracy than the baseline and the two regularization approaches across all tested angles.

Table 5: Two moons data. Target error rates for different rotation angles, obtained by training an SVM classifier with a Gaussian kernel on the transported source data.

| rotation angle | Diffusion-OT | OT | OT-reg | OT-Laplace |
|---|---|---|---|---|
| $10°$ | 0 | 0 | 0 | 0 |
| $20°$ | 0 | 0 | 0 | 0 |
| $30°$ | 0 | 0 | 0 | 0.02 |
| $40°$ | 0 | 0.05 | 0.09 | 0.14 |
| $50°$ | 0 | 0.15 | 0.14 | 0.16 |
| $60°$ | 0.08 | 0.19 | 0.2 | 0.21 |
| $70°$ | 0.21 | 0.26 | 0.27 | 0.31 |
| $80°$ | 0.3 | 0.33 | 0.34 | 0.36 |
| $90°$ | 0.35 | 0.38 | 0.38 | 0.42 |

## D.2 DEEP DOMAIN ADAPTATION

In the following section, we outline the framework utilized in the experiments presented in Section 5.2. For these experiments, we adopt the framework introduced by Damodaran et al. (2018). This framework, termed DeepJDOT, comprises two components: an embedding function, denoted by $g : \mathcal{X} \to \mathcal{Z}$, which aims to learn features optimizing both classification accuracy and domain invariance, and a classification function, denoted by $f : \mathcal{Z} \to \mathcal{Y}$, which predicts the labels based on the learned features.

The DeepJDOT objective, which operates on a minibatch of size $m$, is given by:

$$\min_{\gamma,f,g} \sum_{i}^{m} \mathcal{L}(y_i^s, f(g(x_i^s))) + \sum_{i,j}^{m} \gamma_{i,j} \left( \eta_1 \|g(x_i^s) - g(x_j^t)\|_2^2 + \eta_2 \mathcal{L}(y_i^s, f(g(x_j^t))) \right), \quad (12)$$

where $\mathcal{L}$ is the cross-entropy loss, $\eta_1$ and $\eta_2$ are hyperparameters, and $\gamma$ is the optimal plan.

The training process follows an alternating optimization scheme. Initially, the parameters of both the feature extractor $g$ and the classifier $f$ are fixed. The fixed-parameter models are denoted as $\hat{f}$ and $\hat{g}$. Subsequently, the OT problem is solved with the associated cost matrix $\mathbf{C}_{i,j} = \eta_1 \|\hat{g}(x_i^s) - \hat{g}(x_j^t)\|_2^2 + \eta_2 \mathcal{L}(y_i^s, \hat{f}(\hat{g}(x_j^t))$. Finally, the optimization proceeds by fixing the obtained plan, $\gamma$, and optimizing the model on the mini-batch using Stochastic Gradient Descent (SGD). The objective function for this training phase is defined by Equation 12.

In our experiments, we propose to derive the plan using the Diffusion-OT cost. We solve the OT problem with the associated cost $\mathbf{C}_{i,j} = -\eta_1 \log(\mathbf{S}_{i,j}) + \eta_2 \mathcal{L}(y_i^s, \hat{f}(\hat{g}(x_j^t)))$, where:

$$\mathbf{S}_{i,j} = \sum_{k|y_k^s=y_i^s} \sum_{l \in \mathcal{N}_j^t} \mathbf{P}_s(i,k) \mathbf{Q}(k,l) \mathbf{P}_t(l,j). \tag{13}$$

The probability matrices $\mathbf{P}_s$, $\mathbf{Q}$ and $\mathbf{P}_t$, are obtained by applying a row-stochastic normalization to the kernels defined in Equations 11, 3 and 8, respectively. However, in this case, the kernels are applied in the latent space, using the source and target representations $\{\hat{g}(x_i^s)\}_{i=1}^m$ and $\{\hat{g}(x_j^t)\}_{j=1}^m$, where $m$ denotes the batch size. For the source and target kernels, we compute the kernels by employing the Cosine distance, while for the cross-domain we use the Euclidean distance. $\mathcal{N}_j^t$ denotes the neighborhood of $x_j^t$, determined by the three nearest neighbors in the features space. $\epsilon_s, \epsilon_c$ and $\epsilon_t$ are scale hyper-parameters. In all deep experiments, we set $\epsilon_s = \epsilon_c = \epsilon_t = 1$.

Once the plan is obtained, we fix $\hat{\gamma}$ and train $f$ and $g$ using Equation 12. We note that for simplicity in handling derivatives, we omit the use of Diffusion-OT in the second phase of training, and utilize Equation 12 as is, although integrating the Diffusion-OT cost remains a viable option.

**Datasets.** The **Digits** dataset comprises three widely-used digit datasets, serving as the different domains: (i) MNIST (28x28 grayscale images of handwritten digits), (ii) USPS (16x16 grayscale images of digits derived from scanned handwritten letters), and (iii) SVHN (32x32 colored images of digits with diverse backgrounds and fonts collected from Google Street View). All datasets consist of 10 classes, corresponding to the digits 0 through 9. The **Office-Home** dataset is composed of images from four distinct domains, each representing a specific visual style and content distribution. The four domains are Art (A), Clipart (C), Product (P), and Real-World (R), all with 65 classes. The **VisDA-2017** dataset comprises synthetic and real-world images across 12 diverse classes and often serves as a benchmark for domain adaptation in computer vision. In our experimental setup, we designate the training set containing synthetic images as the source domain, while the validation set, consisting of real images, serves as the target domain.

### D.2.1 DIGITS

For the digits experiments, we adopt the architecture proposed by Damodaran et al. (2018). The generator $g$ is a CNN model trained from scratch, with convolutional layers containing 32, 32, 64, 64, 128, and 128 filters, followed by a fully connected (FC) layer of 128 hidden units and a Sigmoid activation function. Layer normalization is applied after each convolution layer. The classifier $f$ is an FC layer with 10 units, corresponding to the number of classes. We maintain a batch size of $m = 500$ for both the source and target datasets. We construct balanced batches for the source training set by utilizing the labels, ensuring an equal number of samples for each class. The models are optimized using the Adam optimizer. Initially, the training is performed on the source data using cross-entropy loss for 10 epochs. Subsequently, the models are trained for 100 epochs using both the source and target data, with the objective as described above.

For the proposed method, Diffusion-UOT, we use a learning rate $lr = 0.001$. We set the hyperparameters $\eta_1 = 1$, $\eta_2 = 1$ for the objective function, $\epsilon = 1$ for all the Gaussian kernels constructing our diffusion operator, and $\lambda = 0.02$, $\tau = 1$ for the unbalanced Sinkhorn problem. For the reproduced methods we use the parameters reported in the original papers. For all reproduced methods, we use a learning rate $lr = 0.0002$. For DeepJDOT, we use $\eta_1 = 0.001$, $\eta_2 = 0.0001$. For JUM-BOT, we use $\eta_1 = 0.1$, $\eta_2 = 0.1$ for the objective function, and $\lambda = 0.1$, $\tau = 1$ for the unbalanced Sinkhorn problem. We note that since we struggled to reproduce the m-POT results for this specific dataset, Table 1a presents the m-POT results reported in the original paper (Nguyen et al., 2022a). We ran each method 3 times, Table 6 presents the results for each run.

Table 6: Accuracy for Digits dataset over three seeds for (a) DeepJDOT (b) JUMBOT and (c) Diffusion-UOT (ours).

(a)

| Run | SVHN-MNIST | USPS-MNIST | MNIST-USPS |
|-----|------------|------------|------------|
| 1 | 95.81 | 96.96 | 86.05 |
| 2 | 95.53 | 97.34 | 86.75 |
| 3 | 96.78 | 97.36 | 85.55 |

(b)

| Run | SVHN-MNIST | USPS-MNIST | MNIST-USPS |
|-----|------------|------------|------------|
| 1 | 99.05 | 98.64 | 96.66 |
| 2 | 98.99 | 98.62 | 97.41 |
| 3 | 98.93 | 98.77 | 96.71 |

(c)

| Run | SVHN-MNIST | USPS-MNIST | MNIST-USPS |
|-----|------------|------------|------------|
| 1 | 99.15 | 98.90 | 97.81 |
| 2 | 99.23 | 98.87 | 98.11 |
| 3 | 99.19 | 98.84 | 97.51 |

### D.2.2 OFFICE-HOME

For this dataset, the generator $g$ is a pre-trained ResNet50 with the final FC layer removed, and the classifier $f$ consists of an FC layer. We construct balanced batches for the source training set by utilizing the labels, ensuring an equal number of samples for each class. The models are optimized using the SGD optimizer with a learning rate of $0.001$. Given that our cost incorporates source neighborhood information based on source labels, it is required for each mini-batch to contain at least two samples from every class. Notably, this dataset comprises 65 classes. Thus, we employ a batch size of $m = 195$, and train the model for 4000 iterations. For all reproduced methods, we follow the original papers by using a batch size of $m = 65$ and training the model for 10000 iterations.

For the proposed method, Diffusion-UOT, we set the hyperparameters $\eta_1 = 0.01$, $\eta_2 = 2$. for the objective function, $\epsilon = 1$ for all the Gaussian kernels constructing our diffusion operator, and $\lambda = 0.02$, $\tau = 0.5$ for the unbalanced Sinkhorn problem. For the reproduced methods, we use the parameters reported in the original papers. For DeepJDOT, we set $\eta_1 = 0.01$ and $\eta_2 = 0.05$ for the objective function. For JUMBOT and TS-POT, we use $\eta_1 = 0.01$, $\eta_2 = 0.5$ for the objective function. For JUMBOT, we use $\lambda = 0.01$, $\tau = 0.5$ for the unbalanced Sinkhorn problem. For TS-POT, we set the fraction of mass to $0.6$, and the number of mini-batches $k$ to 2.

### D.2.3 VISDA

Similar to the approach taken with the Office-Home dataset, we utilize a pre-trained ResNet50 as the generator $g$, with the classifier $f$ implemented as an FC layer. For all methods, we use a batch size $m = 72$. We construct balanced batches for the source training set by utilizing the labels, ensuring an equal number of samples for each class. The models are optimized using the SGD optimizer with a learning rate of $0.0005$, and trained for 10000 iterations. We employ the two-stage (TS) implementation proposed by Nguyen et al. (2022a) for our method and for TS-POT. In this implementation, the OT problem is initially solved using a batch size of $k \times m$ on the CPU, leveraging its ability to handle larger matrices. After obtaining the optimal plan, the gradient step is executed on the GPU with a batch size of $m$. To utilize the large plan in loss functions computed on smaller batch sizes, an adaptation to the original loss function is proposed. For more details, see Nguyen et al. (2022a).

For the proposed method, TS-Diffusion-POT, we set the hyperparameters $\eta_1 = 0.01$, $\eta_2 = 1$ for the objective function, $\epsilon = 1$ for all the Gaussian kernels constructing our diffusion operator. For the

Table 7: Accuracy for Office-Home dataset over three seeds for (a) DeepJDOT (b) JUMBOT (c) TS-POT and (d) Diffusion-UOT (ours).

(a)

| Run | A-C | A-P | A-R | C-A | C-P | C-R | P-A | P-C | P-R | R-A | R-C | R-P | Avg |
|-----|-----|-----|-----|-----|-----|-----|-----|-----|-----|-----|-----|-----|-----|
| 1 | 51.8 | 70.7 | 76.4 | 59.7 | 66.9 | 69.1 | 57.6 | 49.1 | 75.3 | 69.4 | 54.6 | 80.1 | 65.07 |
| 2 | 52.2 | 71.1 | 76.2 | 60.8 | 67.2 | 69.5 | 58.8 | 48.4 | 75.4 | 68.7 | 54.8 | 79.8 | 65.22 |
| 3 | 51.9 | 70.9 | 75.8 | 60.9 | 65.9 | 68.9 | 58.8 | 48.7 | 75.2 | 68.6 | 55.3 | 79.8 | 65.06 |

(b)

| Run | A-C | A-P | A-R | C-A | C-P | C-R | P-A | P-C | P-R | R-A | R-C | R-P | Avg |
|-----|-----|-----|-----|-----|-----|-----|-----|-----|-----|-----|-----|-----|-----|
| 1 | 55.7 | 75.0 | 80.7 | 65.3 | 74.3 | 75.1 | 65.3 | 53.1 | 79.5 | 74.6 | 59.3 | 83.8 | 70.16 |
| 2 | 56.0 | 74.6 | 80.7 | 64.4 | 74.3 | 74.9 | 65.5 | 53.4 | 79.6 | 74.5 | 59.4 | 83.9 | 70.11 |
| 3 | 55.5 | 75.4 | 80.6 | 65.5 | 74.9 | 75.1 | 65.2 | 53.3 | 79.7 | 74.4 | 59.2 | 84.0 | 70.23 |

(c)

| Run | A-C | A-P | A-R | C-A | C-P | C-R | P-A | P-C | P-R | R-A | R-C | R-P | Avg |
|-----|-----|-----|-----|-----|-----|-----|-----|-----|-----|-----|-----|-----|-----|
| 1 | 57.7 | 76.3 | 81.7 | 68.4 | 73.7 | 76.9 | 67.3 | 55.2 | 80.6 | 75.4 | 60.2 | 83.7 | 71.43 |
| 2 | 58.4 | 77.4 | 81.4 | 67.9 | 72.5 | 76.2 | 67.4 | 54.8 | 80.7 | 75.4 | 59.1 | 84.2 | 71.29 |
| 3 | 56.3 | 77.5 | 81.7 | 68.7 | 72.2 | 76.5 | 67.7 | 55.3 | 80.4 | 75.5 | 60.4 | 84.1 | 71.36 |

(d)

| Run | A-C | A-P | A-R | C-A | C-P | C-R | P-A | P-C | P-R | R-A | R-C | R-P | Avg |
|-----|-----|-----|-----|-----|-----|-----|-----|-----|-----|-----|-----|-----|-----|
| 1 | 57.0 | 77.2 | 82.4 | 69.8 | 74.8 | 79.4 | 68.0 | 56.7 | 82.2 | 75.3 | 60.5 | 84.6 | 72.33 |
| 2 | 57.0 | 78.3 | 82.0 | 70.3 | 74.9 | 78.4 | 68.2 | 56.4 | 82.0 | 75.7 | 61.2 | 84.5 | 72.40 |
| 3 | 57.6 | 78.4 | 81.8 | 70.6 | 74.9 | 78.7 | 68.2 | 56.3 | 82.0 | 75.8 | 61.0 | 85.2 | 72.55 |

TS implementation, we set $s = 0.5$ as the fraction of mass for the partial OT problem, and $k = 2$ as the number of mini-batches. For the reproduced methods, we use the parameters reported in the original papers. For DeepJDOT, we use $\eta_1 = 0.005$ and $\eta_2 = 0.1$ for the objective function. For JUMBOT and TS-POT, we use $\eta_1 = 0.005$ and $\eta_2 = 1$ for the objective function. For JUMBOT, we use $\lambda = 0.01$, $\tau = 0.3$ for the unbalanced Sinkhorn problem. For TS-POT, we set the fraction of mass to 0.75, and $k = 1$ for the TS implementation. In Tables 8, we present the results per class for TS-Diffusion-POT and the competing methods.

Additionally, we analyze the t-SNE representation of the VisDA target features obtained from the deep model trained with Diffusion-OT, compared to those learned using TS-POT (Nguyen et al., 2022a), the baseline for this dataset. For better visualization, we randomly sampled $10\%$ of the samples from each class before applying t-SNE.

While the deep model is designed to learn representations that are both domain-invariant and op-timized for source classification accuracy, resulting in well-separated classes for both methods (as shown in the figure), the visualization still highlights differences between the methods. These dif-ferences help explain why Diffusion-OT achieves better target accuracy compared to the standard transportation cost. For example, in Figure 5a, which shows the TS-POT features, the "knife" class (colored magenta and circled by a red dashed line) completely overlaps with the "skateboard" class (colored black). In contrast, Figure 5b, which presents features obtained using TS-Diffusion-POT, shows that these classes are well-separated. Additionally, the TS-POT model appears to have learned a representation with more than the expected 12 clusters. For instance, the cluster circled by a gold dashed line in Figure 5a does not correspond to any specific class. In contrast, while TS-Diffusion-POT does not achieve perfect separation between classes (as expected, given the approximately $78.5\%$ target accuracy), the label associated with each cluster is easily identifiable.

Table 8: Accuracy for VisDA dataset per class over three seeds for (a) DeepJDOT (b) JUMBOT (c) TS-POT and (d) TS-Diffusion-POT (ours).

(a)

| Run | Avg | plane | bicycle | bus | car | horse | knife | mcycle | person | plant | sktbrd | train | truck | $\text{Avg}_{\text{class}}$ |
|---|---|---|---|---|---|---|---|---|---|---|---|---|---|---|
| 1 | 69.66 | 88.6 | 59.2 | 70.1 | 67.2 | 87.6 | 4.0 | 90.4 | 68.3 | 93.5 | 49.6 | 88.4 | 31.2 | 66.5 |
| 2 | 69.88 | 91.3 | 57.8 | 69.0 | 72.5 | 84.5 | 1.9 | 89.9 | 61.1 | 91.9 | 64.2 | 84.9 | 30.5 | 66.6 |
| 3 | 69.21 | 87.8 | 57.1 | 70.9 | 69.0 | 87.2 | 3.2 | 91.7 | 70.2 | 92.3 | 46.8 | 87.5 | 25.2 | 65.7 |

(b)

| Run | Avg | plane | bicycle | bus | car | horse | knife | mcycle | person | plant | sktbrd | train | truck | $\text{Avg}_{\text{class}}$ |
|---|---|---|---|---|---|---|---|---|---|---|---|---|---|---|
| 1 | 72.84 | 93.0 | 53.4 | 76.9 | 76.6 | 89.8 | 2.1 | 94.5 | 76.2 | 94.4 | 71.2 | 89.7 | 18.4 | 69.7 |
| 2 | 72.81 | 93.7 | 56.9 | 75.5 | 71.8 | 90.5 | 3.5 | 94.2 | 77.6 | 96.1 | 60.6 | 89.0 | 27.4 | 69.7 |
| 3 | 73.27 | 89.7 | 57.0 | 74.8 | 77.3 | 90.6 | 1.5 | 93.9 | 74.6 | 94.4 | 69.9 | 90.3 | 24.6 | 69.9 |

(c)

| Run | Avg | plane | bicycle | bus | car | horse | knife | mcycle | person | plant | sktbrd | train | truck | $\text{Avg}_{\text{class}}$ |
|---|---|---|---|---|---|---|---|---|---|---|---|---|---|---|
| 1 | 75.08 | 94.4 | 66.6 | 78.8 | 73.2 | 93.4 | 1.9 | 95.2 | 73.9 | 94.8 | 79.8 | 88.4 | 31.6 | 72.7 |
| 2 | 76.54 | 95.1 | 61.4 | 82.1 | 75.2 | 91.9 | 67.1 | 94.8 | 79.2 | 96.2 | 80.0 | 89.8 | 13.7 | 77.2 |
| 3 | 75.32 | 93.1 | 68.8 | 81.6 | 74.7 | 93.8 | 2.1 | 94.4 | 75.1 | 95.9 | 76.7 | 90.8 | 26.4 | 72.8 |

(d)

| Run | Avg | plane | bicycle | bus | car | horse | knife | mcycle | person | plant | sktbrd | train | truck | $\text{Avg}_{\text{class}}$ |
|---|---|---|---|---|---|---|---|---|---|---|---|---|---|---|
| 1 | 78.48 | 95.0 | 58.6 | 82.3 | 76.9 | 95.5 | 67.6 | 95.1 | 79.2 | 95.3 | 84.2 | 89.7 | 26.9 | 78.9 |
| 2 | 78.74 | 95.0 | 64.3 | 85.2 | 77.2 | 95.5 | 68.7 | 94.0 | 78.6 | 94.7 | 82.8 | 88.9 | 25.7 | 79.2 |
| 3 | 78.47 | 94.8 | 61.7 | 85.5 | 75.9 | 95.7 | 70.8 | 94.1 | 79.3 | 95.2 | 82.8 | 87.8 | 25.9 | 79.1 |

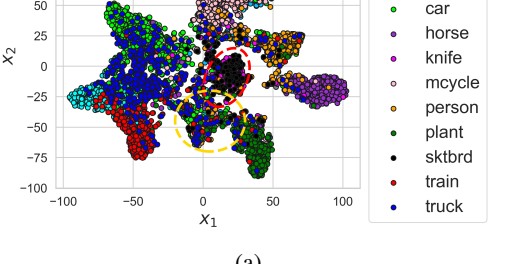

(a)

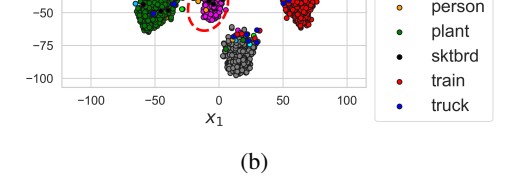

(b)

Figure 5: t-SNE visualization of VisDA features, learned from (a) the TS-POT model and (b) the TS-Diffusion-POT model (ours).

### D.3    DOMAIN ADAPTATION FOR NON-EUCLIDEAN DATA

In the following section, we outline the framework utilized in the experiments presented in Section 5.3.

Previous studies (Barachant et al., 2011; 2013; Zanini et al., 2017; Rodrigues et al., 2018) have demonstrated the efficacy of using the empirical covariance matrices of the EEG recordings as an informative feature representation for this type of data. Following this established approach, we adhere to the framework introduced by Yair et al. (2019), which solves the DA problem using OT on the (non-Euclidean) Riemannian manifold of Symmetric Positive Definite (SPD) matrices.

Consider the source data samples and labels as $\{(\mathbf{X}_i^s, y_i^s)\}_{i=1}^{N_s}$ and the target data samples as $\{\mathbf{X}_j^t\}_{j=1}^{N_t}$. Each data sample $\mathbf{X}_i$ is a covariance matrix, which is an SPD matrix that lies on the Riemannian manifold of SPD matrices $\mathcal{M} \subset \mathbb{R}^{d \times d}$, where $d$ is the number of EEG channels. Applying Algorithm 1 with the Log-Euclidean metric (Arsigny et al., 2006), we derive the Diffusion-OT cost. Next, we solve the OT problem using the Sinkhorn algorithm (Cuturi, 2013), as described in Section 3. Subsequently, leveraging the obtained optimal plan $\gamma$, we map the source features into the target domain. This mapping can be computed using the barycentric mapping (Courty et al., 2016; Perrot et al., 2016), as described in Section 3.

When utilizing the Log-Euclidean metric to compute the cost function, the barycenter can be represented as the weighted Riemannian mean, expressed by the following closed-form:

$$\mathbf{Z}_i^s = \exp\left(\sum_{j=1}^{N_t} \gamma_{i,j} \log\left(\mathbf{X}_j^t\right)\right), \tag{14}$$

where $\exp(\cdot)$ and $\log(\cdot)$ are the matrix exponential and logarithm. The framework is summarized in Algorithm 2.

---

**Algorithm 2** Diffusion-OT on the Riemannian manifold of SPD matrices

---

**Input**: $\{(\mathbf{X}_i^s, y_i^s)\}_{i=1}^{N_s}, \{\mathbf{X}_j^t\}_{j=1}^{N_t}, \epsilon_s, \epsilon_c, \epsilon_t$.

1: Compute the Diffusion-OT cost $\mathbf{C}$ by applying Algorithm 1 to the source and target matrices.
2: Obtain the optimal plan $\gamma$ by applying the Sinkhorn algorithm.
3: Compute the barycentric mappings of the source samples using equation 14.
4: Return the source mappings.

---

Since the representations, $\{\mathbf{Z}_i^s\}_{i=1}^{N_s}$ and $\{\mathbf{X}_j^t\}_{j=1}^{N_t}$, are covariance matrices that lie on the Riemannian manifold of SPD matrices $\mathcal{M}$, prior to conducting any linear computation, such as training a linear classifier, we project both the source and target covariance matrices onto the tangent space $\mathcal{T}_{\mathbf{M}}\mathcal{M}$, using the logarithmic map:

$$\text{Log}_{\mathbf{M}}(\mathbf{Z}_i) = \log(\mathbf{Z}_i) - \log(\mathbf{M}), \tag{15}$$

where $\mathbf{M}$ represents the Riemannian mean of all the covariance matrices.

**Datasets.** The MI1 dataset (Blankertz et al., 2007) contains EEG recordings from 7 subjects utilizing 59 electrodes. Subjects were instructed to perform two out of three MI tasks: imagining the movement of the left hand, the right hand, or the foot. Our evaluation focuses exclusively on the calibration data, comprising 100 trials for each subject. EEG signals were recorded at 1000 Hz, bandpass-filtered between 0.05 Hz and 200 Hz, and down-sampled to 100 Hz. The MI2 dataset (Tangermann et al., 2012) consists of EEG recordings from 9 subjects using 22 electrodes. The experiments include four MI tasks: the imagination of moving the left hand (class 1), right hand (class 2), both feet (class 3), and tongue (class 4). The recordings in MI2 of each subject contain two sessions, recorded on separate days, with 72 trials conducted for each MI task in each session, resulting in 288 trials per session. EEG signals were sampled at 250 Hz and bandpass-filtered between 0.5 Hz and 100 Hz. In both datasets, subjects were presented with a cue at each trial, instructing them to imagine the corresponding MI task.

**Pre-processing.** We apply the same pre-processing pipeline proposed in Zhang & Wu (2020) to both datasets. Initially, we extract a 3-second time window, spanning from $t = 0.5s$ to $t = 3.5s$ following the cue. Next, a bandpass (BP) filter ranging from 8 Hz to 30 Hz is applied to the data. Following this, we compute the empirical covariance matrix of each trial. Finally, centroid alignment (CA) (Zanini et al., 2017; Zhang & Wu, 2020) is applied to each set of covariance matrices, representing a domain, according to:

$$\mathbf{X}_i^{(k)} = \left(\mathbf{M}^{(k)}\right)^{-\frac{1}{2}} \widetilde{\mathbf{X}}_i^{(k)} \left(\mathbf{M}^{(k)}\right)^{-\frac{1}{2}}, \tag{16}$$

where $\widetilde{\mathbf{X}}_i^{(k)}$ denotes the empirical covariance of trial $i$ of the $k$-th subject, and $\mathbf{M}^{(k)}$ represents the Riemannian mean of the collection of covariance matrices of the $k$-th subject. The pre-processed set of each subject is denoted by $\{\mathbf{X}_i^{(k)}\}_{i=1}^{N}$, where $N$ is the total number of trials, possibly with the set of corresponding class labels $\{y_i\}_{i=1}^{N}$.

### D.3.1 BINARY CLASSIFICATION

**Single-source domain adaptation.** In the cross-subject experiment, we apply Algorithm 2, where each subject, in turn, serves as the target domain, with the remaining subjects alternately acting as the source domain. Accuracy is computed by initially mapping the obtained source and target representations onto a tangent space, following Equation 15, and subsequently training a linear SVM classifier on the transformed source mappings and evaluating it on the target mappings. The presented results showcase the average target accuracy across source subjects.

For the cross-subject task, we employed the Sinkhorn algorithm with entropy regularization $\lambda = 0.02$. For the first dataset, MI1, we used the following scale parameters for the probability kernels to produce the results shown in Table 3a: $\epsilon_s = 0.4$, $\epsilon_c = 1$, and $\epsilon_t = 1$. For the second dataset, MI2, the scale parameters for the probability kernels were set as follows: $\epsilon_s = 0.9$, $\epsilon_c = 2$, and $\epsilon_t = 2$.

In Tables 9 and 10, we present the accuracy per subject on this task, including a comparison with the SOTA method, MEKT (Zhang & Wu, 2020), utilizing their provided source code for our analysis.

Table 9: Binary classification accuracy for the cross-subject task on MI1, presented per subject.

|  | 1 | 2 | 3 | 4 | 5 | 6 | 7 | Avg | STD |
|---|---|---|---|---|---|---|---|---|---|
| MEKT-R | **71.58** | **65.58** | **64.25** | 65.58 | 83.33 | 64.67 | **81.92** | 70.99 | 8.33 |
| Diffusion-OT | 70.92 | 63.5 | 63.25 | **69.75** | **86.33** | **72.67** | 79.25 | **72.24** | 8.3 |
| Difference | -0.66 | -2.08 | -1.0 | 4.17 | 3.0 | 8.0 | -2.67 | 1.25 | 3.92 |

Table 10: Binary classification accuracy for the cross-subject task on MI2, presented per subject.

|  | 1 | 2 | 3 | 4 | 5 | 6 | 7 | 8 | 9 | Avg | STD |
|---|---|---|---|---|---|---|---|---|---|---|---|
| MEKT-R | 74.57 | 50.69 | **84.55** | 64.06 | 53.39 | 64.06 | 59.38 | 86.55 | 74.31 | 67.95 | 12.84 |
| Diffusion-OT | **75.52** | **53.39** | 83.94 | **67.27** | **57.55** | **65.02** | **63.8** | **91.06** | **76.13** | **70.41** | 12.28 |
| Difference | 0.95 | 2.7 | -0.61 | 3.21 | 4.16 | 0.96 | 4.42 | 4.51 | 1.82 | 2.46 | 1.8 |

**Multi-source domain adaptation.** For the leave-one-out task, we utilize the Wasserstein Barycenter Transport (WBT) framework, proposed in Montesuma & Mboula (2021). This method involves transporting all source samples to an intermediate domain, termed Wasserstein Barycenter Transport (WBT), before subsequently transferring the source samples from WBT to the target domain using standard OTDA. For the transportation of the source samples to the WBT only, we utilized the label-enhanced kernel in Equation 11 for both the source domain and the WBT, leveraging the available labels. For simplicity, we initially map both the source and target covariance matrices onto a tangent space, using Equation 15, and then apply the WBT algorithm with the Euclidean metric. Notably, we utilize Diffusion-OT instead of the standard cost for all OT applications within the WBT framework, including transportation from each source domain to WBT and from the Wasserstein barycenter of sources to the target domain. The results for these experiments appear in Table 3b.

For both datasets, we set $\lambda^b = 0.02$ for transporting all the source domains to the WBT and $\lambda = 0.1$ for transporting samples from the WBT to the target domain. In the latter case, we apply max normalization to the cost matrix before using the Sinkhorn algorithm. For the first dataset, MI1, the scale parameters for the probability kernels used to transport all source domains to the WBT are $\epsilon_s^b = 0.05$, $\epsilon_c^b = 0.3$, and $\epsilon_t^b = 0.05$. For the transportation from the WBT to the target domain, the parameters are $\epsilon_s = 0.05$, $\epsilon_c = 0.02$, and $\epsilon_t = 0.03$. For the second dataset, MI2, the scale parameters for the probability kernels used to transport all source domains to the WBT are $\epsilon_s^b = 0.01$, $\epsilon_c^b = 0.3$, and $\epsilon_t^b = 0.01$. For the transportation from the WBT to the target domain, the parameters are $\epsilon_s = 0.3$, $\epsilon_c = 0.02$, and $\epsilon_t = 0.02$.

In Tables 11 and 12, we present the accuracy per subject on this task, including a comparison with the SOTA method, MEKT (Zhang & Wu, 2020).

Table 11: Binary classification accuracy for the leave-one-out task on MI1, presented per subject.

| | 1 | 2 | 3 | 4 | 5 | 6 | 7 | Avg | STD |
|---|---|---|---|---|---|---|---|---|---|
| MEKT-R | 86.5 | 69.5 | 73.5 | 89.0 | 94.0 | **89.5** | 91.5 | 84.79 | 9.43 |
| Diffusion-OT | **90.0** | **78.0** | **75.0** | **92.0** | **96.0** | 87.0 | **93.0** | **87.29** | 7.91 |
| Difference | 3.5 | 8.5 | 1.5 | 3.0 | 2.0 | -2.5 | 1.5 | 2.5 | 3.28 |

Table 12: Binary classification accuracy for the leave-one-out task on MI2, presented per subject.

| | 1 | 2 | 3 | 4 | 5 | 6 | 7 | 8 | 9 | Avg | STD |
|---|---|---|---|---|---|---|---|---|---|---|---|
| MEKT-R | **93.06** | 49.31 | 95.83 | 73.61 | 56.25 | **70.83** | **70.14** | 94.44 | **83.33** | 76.31 | 16.76 |
| Diffusion-OT | 91.67 | **51.39** | **97.92** | **75.0** | **61.81** | 70.14 | 68.06 | **96.53** | 82.64 | **77.24** | 16.14 |
| Difference | -1.39 | 2.08 | 2.09 | 1.39 | 5.56 | -0.69 | -2.08 | 2.09 | -0.69 | 0.93 | 2.38 |

### D.3.2 MULTI-CLASS CLASSIFICATION

In this section, we assess the performance of Diffusion-OT in a multi-class classification task using the MI2 dataset, taking into account all four classes available in the dataset. In addition to the baseline methods, OT and OT-reg (Courty et al., 2016; 2014; Yair et al., 2019), we compare Diffusion-OT to standard DA methods, including CORAL (Sun et al., 2017), SA (Fernando et al., 2013), and TCA (Pan et al., 2010).

For all the multi-class classification experiments we employ the Sinkhorn algorithm with entropy regularization parameter $\lambda = 0.1$ and apply Algorithm 1 with $\epsilon_s = 3, \epsilon_c = 0.1, \epsilon_t = 0.1$. For the reproduced methods, OT and OT-reg, we apply median normalization to the cost matrix before applying the Sinkhorn algorithm. We set $\lambda = 0.005$ for the entropy parameter. For OT-reg, we use $\eta = 10$ for the label regularization, as suggested in the original paper (Yair et al., 2019). For CORAL, SA, and TCA, we utilize the ADAPT package (de Mathelin et al., 2021) with the default parameters.

Table 13a shows the results for the cross-session task. In this experiment, for each subject, the first session (day) serves as the source domain, while the second session serves as the target domain, and vice versa. The reported target accuracy represents the average accuracy across both sessions, obtained as follows: First, we apply Algorithm 2 to obtain the transported source samples. Next, we map both the target and the transported source covariance matrices to a tangent plane, as described in Equation 15. We then train a linear SVM on the mapped source samples and test it on the target samples. Following Zanini et al. (2017); Yair et al. (2019), we present results for five subjects out of the available nine, as the remaining subjects exhibited poor results and were considered invalid in those works. We observe that the proposed method achieves the highest accuracy for three out of the five subjects. For subject 1, however, it shows lower accuracy compared to the baseline OT-reg. This is the only instance across all experiments where we do not achieve an improvement over the baseline. Notably, Diffusion-OT yields the highest average accuracy, surpassing OT and OT-reg by over 6% and 2%, respectively. Detailed results for all subjects appear in Table 14.

In Table 13b, we present the results for the cross-subject task, analogous to the results shown in Table 3a for the binary classification case. Same as in the cross-session task, only the five valid subjects are considered. Our method achieves the highest accuracy across all subjects and demonstrates an average performance that significantly exceeds both baseline methods Detailed results for all nine subjects can be found in Table 15.

## E PROOFS

**Lemma 1.** *The asymptotic expansion of a single-domain diffusion operator, defined in Equation 5, is expressed as follows:*

$$P_\epsilon f(x) = f(x) - \frac{m_2}{m_0}\epsilon \left( \Delta f - f\frac{\Delta \mu}{\mu} \right)(x) + O(\epsilon^2). \tag{17}$$

Table 13: Multi-class classification accuracy on dataset MI2. (a) Results for the cross-session task. (b) Results for the cross-subject task.

(a)

| Method | 1 | 3 | 7 | 8 | 9 | Avg |
|---|---|---|---|---|---|---|
| CORAL | 82.3 | 87.7 | 80.9 | 83.0 | 72.9 | 81.35 |
| SA | 83.7 | **87.8** | 81.4 | 83.9 | 74.8 | 82.33 |
| TCA | 71.0 | 78.1 | 64.6 | 77.8 | 70.1 | 72.33 |
| OT | 84.0 | 85.9 | 81.1 | 76.4 | 68.2 | 79.13 |
| OT-reg | **85.2** | 84.5 | 81.1 | 85.4 | 78.1 | 82.88 |
| Ours | 84.7 | 86.5 | **84.5** | **88.9** | **81.4** | **85.21** |

(b)

| Method | 1 | 3 | 7 | 8 | 9 | Avg |
|---|---|---|---|---|---|---|
| CORAL | 53.1 | 58.2 | 52.0 | 52.5 | 53.3 | 53.82 |
| SA | 58.6 | 63.2 | 53.8 | 59.3 | 58.2 | 58.63 |
| TCA | 59.1 | 61.2 | 49.4 | 60.7 | 52.7 | 56.64 |
| OT | 57.2 | 65.4 | 59.7 | 57.5 | 52.7 | 58.51 |
| OT-reg | 63.2 | 69.7 | 63.8 | 66.1 | 62.1 | 64.97 |
| Ours | **66.7** | **75.1** | **69.4** | **73.4** | **65.5** | **70.03** |

Table 14: Comparison of cross-session multi-class classification on the MI2 dataset.

| Subject | CORAL | SA | TCA | OT | OT-reg | Diffusion-OT |
|---|---|---|---|---|---|---|
| 1 | 82.29 | 83.68 | 71.01 | 84.03 | **85.24** | 84.72 |
| 2 | 50.35 | 51.91 | 44.27 | 56.08 | 57.99 | **58.85** |
| 3 | 87.67 | **87.85** | 78.13 | 85.94 | 84.55 | 86.46 |
| 4 | 57.47 | 59.9 | 37.5 | 58.51 | 57.81 | **60.24** |
| 5 | 41.32 | 41.67 | 28.47 | **46.7** | 45.14 | 44.27 |
| 6 | 47.4 | 50.87 | 36.81 | **52.95** | 50.69 | 48.61 |
| 7 | 80.9 | 81.42 | 64.58 | 81.08 | 81.08 | **84.55** |
| 8 | 82.99 | 83.85 | 77.78 | 76.39 | 85.42 | **88.89** |
| 9 | 72.92 | 74.83 | 70.14 | 68.23 | 78.13 | **81.42** |
| Avg | 67.03 | 68.44 | 56.52 | 67.77 | 69.56 | **70.89** |

*Proof of Lemma 1.* According to Lemma 8 from Coifman & Lafon (2006), given an isotropic kernel $k_\epsilon(x, y) = h\left(\frac{\|x-y\|^2}{\epsilon}\right)$ with an exponential decay, and an operator $T_\epsilon$ defined by $T_\epsilon g(x) = \int_\mathcal{M} k_\epsilon(x, y)g(y)dy$, the asymptotic expansion of $T_\epsilon$ for all $g \in \mathcal{C}^3(\mathcal{M})$ and for all $x \in \mathcal{M}$ is given by:

$$T_\epsilon g(x) = m_0 g(x) - m_2 \epsilon \left(\Delta g(x) - w(x)g(x)\right) + O(\epsilon^2), \tag{18}$$

where $\Delta$ is the Laplace–Beltrami operator on $\mathcal{M}$, $m_0, m_2$ are two constants depending on the kernel $h$, and $w(x)$ is a potential function.

For the diffusion operator $P_\epsilon$ defined in Equation 5, we use $g(x) = \frac{f(x)\mu(x)}{d_\epsilon(x)}$, and from Equation 18 we get:

$$P_\epsilon f(x) = m_0 \frac{f(x)\mu(x)}{d_\epsilon(x)} - m_2 \epsilon \left(\Delta\left(\frac{f(x)\mu(x)}{d_\epsilon(x)}\right) - w(x)\frac{f(x)\mu(x)}{d_\epsilon(x)}\right) + O(\epsilon^2), \tag{19}$$

where $d_\epsilon(x) = \int K_\epsilon(x, y)\mu(y)dy$.

Similarly, we use $g(x) = \mu(x)$ in Equation 18, and obtain that the asymptotic expansion of $d_\epsilon$ is given by:

$$d_\epsilon(x) = m_0 \mu(x) - m_2 \epsilon \left(\Delta\mu(x) - w(x)\mu(x)\right) + O(\epsilon^2)$$

$$= m_0 \mu(x)\left(1 - \frac{m_2}{m_0}\epsilon\left(\frac{\Delta\mu}{\mu} - w\right)(x)\right) + O(\epsilon^2) \tag{20}$$

By applying the Taylor expansion of $\frac{1}{1-x}$ and neglecting terms of order $O(\epsilon^2)$ and higher, we obtain:

$$(d_\epsilon)^{-1}(x) = (m_0 \mu(x))^{-1}\left(1 + \frac{m_2}{m_0}\epsilon\left(\frac{\Delta\mu}{\mu} - w\right)(x)\right) + O(\epsilon^2) \tag{21}$$

Table 15: Comparison of cross-subject multi-class classification on the MI2 dataset.

| Subject | CORAL | SA | TCA | OT | OT-reg | Diffusion-OT |
|---------|-------|-----|-----|-----|--------|--------------|
| 1 | 43.66 | 49.57 | 51.91 | 49.1 | 54.24 | **57.99** |
| 2 | 27.19 | 27.82 | 27.58 | 27.79 | **28.23** | 27.44 |
| 3 | 47.34 | 53.34 | 50.59 | 52.89 | 57.7 | **62.08** |
| 4 | 35.31 | 34.71 | 32.82 | 37.7 | 38.85 | **39.29** |
| 5 | 30.96 | 30.57 | 29.42 | 32.79 | 33.26 | **33.51** |
| 6 | 33.18 | 32.54 | 32.08 | 34.13 | 34.09 | **34.42** |
| 7 | 39.69 | 41.41 | 40.82 | 44.95 | 48.11 | **52.21** |
| 8 | 44.67 | 52.07 | 51.81 | 47.73 | 55.53 | **62.04** |
| 9 | 42.76 | 48.67 | 45.5 | 42.45 | 51.68 | **54.94** |
| Avg | 38.31 | 41.19 | 40.28 | 41.06 | 44.63 | **47.1** |

Multiplying Equation 21 by $f$ and subsequently applying the Laplace–Beltrami operator yields:

$$\frac{f\mu}{d_\epsilon}(x) = \frac{f}{m_0}\left(1 + \frac{m_2}{m_0}\epsilon\left(\frac{\Delta\mu}{\mu} - w\right)(x)\right) + O(\epsilon^2) \tag{22}$$

$$\Delta\left(\frac{f\mu}{d_\epsilon}\right)(x) = \frac{\Delta f}{m_0} + \frac{m_2}{m_0^2}\epsilon\Delta\left(f\left(\frac{\Delta\mu}{\mu} - w\right)\right)(x) + O(\epsilon^2) \tag{23}$$

Finally, by substituting Equations 22 and 23 into Equation 19, and neglecting terms of order $O(\epsilon^2)$, we derive the asymptotic expansion of the operator $P_\epsilon$:

$$P_\epsilon f(x) = f(x) - \frac{m_2}{m_0}\epsilon\left(-f\frac{\Delta\mu}{\mu} + wf + \Delta f - wf\right)(x) + O(\epsilon^2)$$

$$= f(x) - \frac{m_2}{m_0}\epsilon\left(\Delta f - f\frac{\Delta\mu}{\mu}\right)(x) + O(\epsilon^2)$$

$\square$

**Lemma 2.** *The asymptotic expansion of the cross-domain diffusion operator, defined by:*

$$Q_\epsilon f(y) = \int \frac{k_\epsilon(x,y)}{d_{t,\epsilon}(x)} f(x)\mu(x)dx, \tag{24}$$

*is expressed as follows:*

$$Q_\epsilon f(x) = f(x)\frac{\mu}{\nu}(x) - \frac{m_2}{m_0}\epsilon\left(\Delta\left(f\frac{\mu}{\nu}\right) - f\frac{\mu}{\nu}\frac{\Delta\nu}{\nu}\right)(x) + O(\epsilon^2). \tag{25}$$

*Proof of Lemma 2.* For the cross-domain diffusion operator defined in Equation 24, with $\epsilon_c$ as the scale hyperparameter of the cross-domain Gaussian kernel, we substitute $g(x) = \frac{f\mu}{d_{t,\epsilon_c}}(x)$ into Equation 18, and obtain:

$$Q_{\epsilon_c}f(x) = m_0\frac{f\mu}{d_{t,\epsilon_c}}(x) - m_2\epsilon_c\left(\Delta\left(\frac{f\mu}{d_{t,\epsilon_c}}\right)(x) - w(x)\frac{f\mu}{d_{t,\epsilon_c}}(x)\right) + O(\epsilon_c^2), \tag{26}$$

where $d_{t,\epsilon_c}(x) = \int k_{\epsilon_c}(x,y)\nu(y)dy$.

Similarly, we substitute $g(x) = \nu(x)$ to Equation 18, and obtain the asymptotic expansion of $d_{t,\epsilon_c}$:

$$d_{t,\epsilon_c}(x) = m_0\nu(x) - m_2\epsilon_c\left(\Delta\nu(x) - w(x)\nu(x)\right) + O(\epsilon_c^2). \tag{27}$$

We utilize the Taylor expansion as in Equation 21, leading to the derivation:

$$(d_{\epsilon_c})^{-1}(x) = (m_0\nu(x))^{-1}\left(1 + \frac{m_2}{m_0}\epsilon_c\left(\frac{\Delta\nu}{\nu} - w\right)(x)\right) + O(\epsilon_c^2). \tag{28}$$

Multiplying Equation 28 by $f$ and subsequently applying the Laplace–Beltrami operator yields:

$$\frac{f\mu}{d_{\epsilon_c}}(x) = \frac{f\mu}{m_0\nu}\left(1 + \frac{m_2}{m_0}\epsilon_c\left(\frac{\Delta\nu}{\nu} - w\right)(x)\right) + O(\epsilon_c^2). \tag{29}$$

$$\Delta\left(\frac{f\mu}{d_{\epsilon_c}}\right)(x) = \frac{1}{m_0}\Delta\left(f\frac{\mu}{\nu}\right) + \frac{m_2}{m_0^2}\epsilon_c\Delta\left(f\frac{\mu}{\nu}\left(\frac{\Delta\nu}{\nu} - w\right)\right)(x). \tag{30}$$

Finally, by substituting Equations 29 and 30 into Equation 26 and neglecting terms of order $O(\epsilon^2)$, we derive the asymptotic expansion of the operator $Q_{\epsilon_c}$:

$$Q_{\epsilon_c}f(x) = f\frac{\mu}{\nu}(x) - \frac{m_2}{m_0}\epsilon_c\left(-f\frac{\mu}{\nu}\frac{\Delta\nu}{\nu} + f\frac{\mu}{\nu}w + \Delta\left(f\frac{\mu}{\nu}\right) - f\frac{\mu}{\nu}w\right)(x) + O(\epsilon_c^2)$$

$$= f\frac{\mu}{\nu}(x) - \frac{m_2}{m_0}\epsilon_c\left(\Delta\left(f\frac{\mu}{\nu}\right) - f\frac{\mu}{\nu}\frac{\Delta\nu}{\nu}\right)(x) + O(\epsilon_c^2).$$

$\square$

**Proposition 1.** Suppose $f \in \mathcal{C}^4(\mathcal{M})$, and suppose $\mu, \nu \in \mathcal{C}^4(\mathcal{M})$ denote the probability measures of the source and target domains, respectively, where $\mu$ is dominated by $\nu$. Denote the Radon–Nikodym (RN) derivative by $\frac{\mu}{\nu}$. For sufficiently small $\epsilon$, the asymptotic expansion of operator $S_\epsilon$ is given by:

$$S_\epsilon f(x) = \frac{\mu}{\nu}(x)\left[f - \frac{m_2}{m_0}\epsilon\left[3\Delta f + 2\left(f\frac{\Delta\left(\frac{\mu}{\nu}\right)}{\frac{\mu}{\nu}} + 2\nabla f\nabla\log\left(\frac{\mu}{\nu}\right)\right)\right. \tag{6}$$

$$\left.\left. - f\left(\frac{\Delta\mu}{\mu} + 2\frac{\Delta\nu}{\nu}\right)\right]\right](x) + O(\epsilon^2),$$

where $\Delta, \nabla$ are the Laplace–Beltrami operator and the covariant derivative on $\mathcal{M}$, respectively, and $m_0, m_2$ are two constants defined by the Gaussian kernel and by the manifold $\mathcal{M}$.

*Proof of Proposition 1.* The proposed diffusion operator is defined by the composition $S_{\underline{\epsilon}}f(x) = P_{t,\epsilon_t}Q_{\epsilon_c}P_{s,\epsilon_s}f(x)$, where $\underline{\epsilon} = (\epsilon_s, \epsilon_c, \epsilon_t)$.

We start by defining the function $g(x) = P_{s,\epsilon_s}f(x)$. Substituting the asymptotic expansion of the source diffusion operator, as derived in Equation 17, into Equation 25, we obtain:

$$Q_{\epsilon_c}P_{s,\epsilon_s}f(x) = \frac{\mu}{\nu}\left(f - \frac{m_2}{m_0}\epsilon_s\left(\Delta f - f\frac{\mu}{\mu}\right)\right)(x) - \frac{m_2}{m_0}\epsilon_c\left(\Delta\left(f\frac{\mu}{\nu}\right) - f\frac{\mu}{\nu}\right)(x) + O(\epsilon^2)$$

$$= f\frac{\mu}{\nu}(x) - \frac{m_2}{m_0}\left(\epsilon_s\frac{\mu}{\nu}\Delta f + \epsilon_c\Delta\left(f\frac{\mu}{\nu}\right) - f\frac{\mu}{\nu}\left(\epsilon_s\frac{\Delta\mu}{\mu} + \epsilon_c\frac{\Delta\nu}{\nu}\right)\right) + O(\epsilon^2), \tag{31}$$

where all terms with order $O(\epsilon^2)$ were neglected.

Remark that assuming $\epsilon_s = \epsilon_c = \epsilon$ yields:

$$Q_\epsilon P_{s,\epsilon}f(x) = f\frac{\mu}{\nu}(x) - \frac{m_2}{m_0}\epsilon\left(\frac{\mu}{\nu}\Delta f + \Delta\left(f\frac{\mu}{\nu}\right) - f\frac{\mu}{\nu}\left(\frac{\Delta\mu}{\mu} + \frac{\Delta\nu}{\nu}\right)\right)(x) + O(\epsilon^2). \tag{32}$$

Next, we define the function $g(x) = Q_{\epsilon_c}P_{s,\epsilon_s}f(x)$, and substitute Equation 32 into the asymptotic expansion of the target diffusion operator, as defined in Equation 17:

$$P_{t,\epsilon_t}Q_{\epsilon_c}P_{s,\epsilon_s}f(x) = f\frac{\mu}{\nu}(x) - \frac{m_2}{m_0}\left(\epsilon_s\frac{\mu}{\nu}\Delta f + \epsilon_c\Delta\left(f\frac{\mu}{\nu}\right) - f\frac{\mu}{\nu}\left(\epsilon_s\frac{\Delta\mu}{\mu} + \epsilon_c\frac{\Delta\nu}{\nu}\right)\right)$$

$$- \frac{m_2}{m_0}\epsilon_t\left(\Delta\left(f\frac{\mu}{\nu}\right) - f\frac{\mu}{\nu}\frac{\nu}{\nu} + f\frac{\mu}{\nu}\frac{\nu}{\nu}\right)(x) + O(\epsilon^2)$$

$$= f\frac{\mu}{\nu}(x) - \frac{m_2}{m_0}\left(\epsilon_s\frac{\mu}{\nu}\Delta f + (\epsilon_c + \epsilon_t)\Delta\left(f\frac{\mu}{\nu}\right)\right.$$

$$\left. - f\frac{\mu}{\nu}\left(\epsilon_s\frac{\Delta\mu}{\mu} + (\epsilon_c + \epsilon_t)\frac{\Delta\nu}{\nu}\right)\right)(x) + O(\epsilon^2), \tag{33}$$

where all terms with order $O(\epsilon^2)$ were neglected.

Assuming $\epsilon_s = \epsilon_c = \epsilon$ we get:

$$P_{t,\epsilon_t}Q_{\epsilon_c}P_{s,\epsilon_s}f(x) = f\frac{\mu}{\nu}(x) - \frac{m_2}{m_0}\epsilon\left(\frac{\mu}{\nu}\Delta f + 2\Delta\left(f\frac{\mu}{\nu}\right) - f\frac{\mu}{\nu}\left(\frac{\Delta\mu}{\mu} + 2\frac{\Delta\nu}{\nu}\right)\right)(x) + O(\epsilon^2).$$
(34)

Finally, by utilizing:

$$\Delta\left(f\frac{\mu}{\nu}\right) = \frac{\mu}{\nu}\Delta f + f\Delta\left(\frac{\mu}{\nu}\right) + 2\nabla f\nabla\left(\frac{\mu}{\nu}\right),$$
(35)

we yield the expression:

$$S_\epsilon f(x) = f\frac{\mu}{\nu}(x) - \frac{m_2}{m_0}\epsilon\left(3\frac{\mu}{\nu}\Delta f + 2f\Delta\left(\frac{\mu}{\nu}\right) - f\frac{\mu}{\nu}\left(\frac{\Delta\mu}{\mu} + 2\frac{\Delta\nu}{\nu}\right) + 4\nabla f\nabla\left(\frac{\mu}{\nu}\right)\right)(x) + O(\epsilon^2).$$
(36)

Lastly, for a more comprehensive analysis, we utilize the property $\frac{\nabla f}{f} = \nabla\log(f)$ to derive the expression presented in Proposition 1. □

