# OpenReview forum: "Diffusion Transportation Cost for Domain Adaptation"
_ICLR.cc/2025/Conference — Submitted to ICLR 2025_

### Official Review · Reviewer_z6YT · 2024-11-03

**Soundness:** 2
**Presentation:** 3
**Contribution:** 2
**Rating:** 6
**Confidence:** 4

**Summary:**

This paper focuses on the methodological aspect of optimal transport (OT) for domain adaptation. A major motivation is that existing OT works rarely consider the construction and impact of cost functions, which is generally crucial for the property of the OT measure and its induced transport plan. To deal with this limitation, this work proposes a diffusion-based cost formulation, which endows the transport cost with the property of cross-distribution propagation. The theoretical result shows that the proposed cross-domain diffusion operator indeed characterizes the cross-domain discrepancy and intra-domain diversity. Experiments are conducted on standard domain adaptation datasets to evaluate the proposed method.

**Strengths:**

+ The motivation for improving OT from the perspective of the cost function is technically sounded.

+ Theoretical results that show the implications of constructed diffusion operator is reasonable.

+ The empirical improvement over other OT-based domain adaptation methods on several benchmarks.

**Weaknesses:**

1. The relation between the diffusion process and OT should be further clarified; besides, it seems that there are fundamental issues in the validity of OT with designed cost.

2. The implications of diffusion operator for distribution shift correction could be improved, e.g., the superiority of proposed cost design is not sufficiently explained in the current manuscript.

3. The details for the optimization procedure could be improved; the comparison experiment for method validation should contain more hard transfer tasks.

**Questions:**

**Concerns**

Q1. As far as I understand the Diffusion-OT, the key difference between it and existing OT works is that Diffusion-OT introduces the composition of stochastic matrices, i.e., ‘source to source transition’, ‘source to target transition’ and ‘target to target transition’, to construct the cost function. However, it seems that there are no explanations on what special properties are ensured by the Diffusion-OT from OT’s view, e.g., JDOT measures the joint distribution discrepancy, POT/UOT relaxes the strict constraints under severe shift scenarios. Some justifications are appreciated.

Q2. Is the constructed cost function $C=-log(S)$ still a metric? Since the metric property is necessary to ensure the validity of OT. Besides, if it is a metric, which kind of discrepancy does it characterize (e.g., joint/conditional/marginal distribution discrepancy)?

Q3. Prop. 1 shows that the diffusion operator $S$ can reflect the cross-domain discrepancy and intra-domain diversity from the view of LB operator. However, it seems that there is no guarantee that the proposed method can control the diffusion process, i.e., suppress cross-domain divergence and enlarge the intra-domain divergence. Therefore, it is hard to understand the learning process and the properties of the proposed method. Detailed discussion on the optimization and learning procedure would be helpful to improve the clarity.

Q4. In the diffusion operator $S$, the three stochastic matrices are construed with distance-based kernel function on original space $\mathcal{X}$, e.g., Eqs. (4)-(5). Should they be formulated in representation space $\mathcal{Z}$? If so, are the representations considered variables under the optimization process? What are the learning principle and its intuitive goal for the diffusion operator $S$?  An in-depth analysis of the diffusion mechanism in the learning process is high expected.

Q5. Though this work achieves improvements over some existing OT methods, the comparison seems insufficient. On the one hand, some advanced OT methods that have similar goals/ideas to the proposed methods are omitted, e.g., key-point guided OT [a], mask OT [b], and general cost function [c]. Especially, considering the label-guided graph construction in Eq. (11), it indeed has the same idea as the mentioned works. On the other hand, the experiment could be extended to larger and harder datasets, e.g., DomainNet.

**References**

[a] Xiang Gu, Yucheng Yang, Wei Zeng, Jian Sun, and Zongben Xu. Keypoint-guided optimal transport with applications in heterogeneous domain adaptation. In NeurIPS, 2022.

[b] Jiying Zhang, Xi Xiao, Long-Kai Huang, Yu Rong, and Yatao Bian. Fine-tuning graph neural networks via graph topology induced optimal transport. In IJCAI, 2022

[c] Asadulaev, A., Korotin, A., Egiazarian, V., Mokrov, P., & Burnaev, E. Neural Optimal Transport with General Cost Functionals. In ICLR, 2024.

---

> ### Author Response · Authors · 2024-11-21
>
> This is comment 1 out of 2.
>
> We thank the reviewer for their insightful questions and for taking the time to carefully review our paper.
> 1. **Key contributions of Diffusion-OT:**
>
>     The key difference between Diffusion-OT and existing methods is that Diffusion-OT modifies the transportation cost itself, whereas methods like JDOT, POT, and UOT focus on proposing new problem formulations.
>     While a few methods, such as ETD and RWOT, also introduce new transport costs, Diffusion-OT captures both cross-domain and intra-domain relationships through a label-guided diffusion process, which is composed of the three stochastic matrices you pointed out.
>     Additionally, in contrast to competing methods, Diffusion-OT is a data-driven approach, meaning its cost can be easily integrated into most OT-based methods, including JDOT, POT, and UOT, as demonstrated in the experiments section.
>     This approach allows OT-based methods to account for data geometries, effectively leverage source label information, and select the problem formulation that best suits the specific DA task, according to the properties you mentioned.
>
> 2. **The validity of OT using Diffusion-OT:**
>
>     Your observation is correct - the Diffusion-OT cost is not a metric.
>     However, in OT, the cost function does not necessarily need to be a metric for the problem to be valid.
>     While the Wasserstein distance requires the cost function to be a metric in order to define a valid distance, OT as a broader framework encompasses more than just the Wasserstein distance.
>     In our case, the proposed cost is not a metric, and the solution to the OT problem does not represent a distance, nor do we require it to be, as we only use the resulting transportation plan for all the DA tasks presented.
>
> 3. **The optimization process:**
>
>     Prop. 1 suggests that the diffusion process accounts for both cross-domain discrepancies and intra-domain geometries.
>     The OT problem seeks to find a transportation plan that minimizes the given cost. In this context, the goal is to minimize domain discrepancies while preserving intra-domain structure. When source labels are available, the diffusion process can be further guided in a task-specific manner.
>
>     In response to your question, we have included an additional illustration in Figure 4 of the supplementary materials, which highlights how the resulting transportation plan is directly influenced by the selected cost.
>     For further details, please refer to Appendix D.1 in the revised version.
>
>     Regarding your fourth question, in the main formulation, we apply the operator to the original space $\mathcal{X}$. However, in the deep experiments, the kernels are computed using the learned features, i.e., in the representation space $\mathcal{Z}$.
>     In this case, the representations are learned using a deep model, and indeed, the model parameters are variables in the optimization process, which follows an alternating optimization framework. In short, we first train a deep model to learn the latent space $\mathcal{Z}$, then fix the model, using the learned features to compute the Diffusion-OT cost and obtain the transportation plan. Subsequently, we incorporate the plan into the loss function and proceed with the next gradient descent step on the deep model, continuing the learning process for the latent space. For further details, please refer to Appendix D.2, which provides an in-depth description of the DeepJDOT framework.

---

> ### Author Response · Authors · 2024-11-21
>
> This is comment 2 out of 2.
>
> 4. **Comparison to additional methods and datasets:**
>
>     In this paper, we specifically focused on the use of OT for unsupervised DA tasks, and as such, our comparison includes only DA methods. However, since there are only a few papers proposing new transport costs, we briefly discussed methods that are outside the DA context in the related work section.
>
>     The method presented in [a] is a semi-supervised approach that assumes the availability of a set of source-target pairs, referred to as key-points. While this method operates under different settings and is therefore not directly comparable, it introduces an interesting approach for computing the transport cost. We have included it in the revised version (see paragraph 3 in the related work section and paragraph 3 in Appendix A). In fact, the idea shares similarities with the approach by Duque et al. (2023), which is included in the related work section and briefly explained in Appendix A.
>
>     In [b], the method is not a DA method, nor does it propose a new cost function for the OT problem, making it outside the scope of this paper.
>
>     In [c], the authors propose a neural network-based framework for learning task-specific cost functionals. However, the two cost functions presented are semi-supervised or supervised, which makes the setting different from ours. Thus, the method will not be included in the comparison tables, but we have added it to the related work section.
>
>     We appreciate bringing these interesting works to our attention, which have helped enrich our literature review.
>
>     As per your suggestion, we plan to add an additional dataset, such as DomainNet, to the experiments section.
>     While obtaining results on a new dataset within the discussion period may not be feasible, we will make every effort to include them in the camera-ready version.

---

> > ### Comment · Reviewer_z6YT · 2024-11-29
> > **Thank you for your responses**
> >
> > I thank the authors for their detailed responses. After carefully checking the rebuttal, the concerns about the clarity of the proposed method are addressed. Overall, I think the basic idea of improving the design of the cost function is sound and the diffusion-based operator is reasonably incorporated, where the experimental results are good. But, there are also several basic issues from methodological aspects, i.e., the metric property for distribution shift correction is lost, and the fundamental merits over existing methodology (though responses provide primary discussion, it seems that the design of general cost metric and label-guided transport mechanism are indeed studied in existing works).
> >
> > Given the rebuttal is generally appropriate, I would like to improve my score to 6 with a borderline positive recommendation. I strongly recommend authors carefully incorporate the responses into revisions with in-depth discussion.

---

### Official Review · Reviewer_k4Z3 · 2024-11-03

**Soundness:** 2
**Presentation:** 2
**Contribution:** 2
**Rating:** 5
**Confidence:** 4

**Summary:**

This paper presents Diffusion-OT, a transport cost designed for the optimal transport (OT) problem with a focus on domain adaptation. Specifically, the authors adopt concepts from manifold learning, i.e., diffusion geometry, to derive an operator that captures intra-domain relationships. This operator quantifies the probability of transporting samples between the source and target domains, forming the foundation of the transportation cost.  Comprehensive theoretical proofs and extensive experiments demonstrate the effectiveness of the proposed method.

**Strengths:**

1.	This paper presents a graph-based transport cost that accounts for both cross-domain distances and intra-domain structures.
2.	Incorporating theoretical concepts from diffusion processes enhances the depth and rigor of the proposed method.
3.	Empirical results demonstrate the effectiveness of Diffusion-OT across various scenarios, indicating its practical applicability.

**Weaknesses:**

1.	The motivation of this paper is unclear. In the Introduction section, the authors state that “one aspect that remains relatively unexplored is the selection of the transportation cost function”, but they fail to provide compelling reasons for the need to investigate this aspect. Moreover, the authors do not analyze the advantages and disadvantages of existing methods, further diminishing the clarity of the research motivation.
2.	The literature review is notably limited, lacking engagement with significant recent advancements in optimal transport and domain adaptation. This paper mainly compares with related works before 2023. It would benefit from a comparison and discussion of additional recent studies, such as [1-5].
3.	Why does the proposed method exhibit inferior performance compared to RWOT on the MNIST-USPS dataset? More discussions are required.
4.	The recent method SPA [2] demonstrates superior performance compared to the proposed methods in the Office-Home (75.3% v.s 72.43%) and VisDA (87.7% v.s 78.56%) datasets. What advantages does the proposed method offer compared to traditional domain adaptation techniques?
5.	The proposed method involves several hyperparameters(e.g., \lamda, \epsilon). Conducting ablation studies on these hyperparameters would provide valuable insights into the sensitivity of the proposed method and its performance.

**Reference**

[1] Probability-Polarized Optimal Transport for Unsupervised Domain Adaptation. AAAI 2024

[2] SPA: A Graph Spectral Alignment Perspective for Domain Adaptation. NeurIPS 2023

[3] COT: Unsupervised Domain Adaptation With Clustering and Optimal Transport. CVPR 2023

[4] Towards Unsupervised Domain Adaptation via Domain-Transformer. IJCV 2024

[5] Prototype-Guided Feature Learning for Unsupervised Domain Adaptation. Pattern Recognition 2023

**Questions:**

Please see above.

---

> ### Author Response · Authors · 2024-11-21
>
> This is comment 1 out of 2.
>
> We thank the reviewer for the thorough review and the valuable feedback.
>
> 1. **Motivation and advantages of Diffusion-OT:**
>
>     Thank you for raising this concern. The statement aims to highlight that while OT is widely explored in DA, most existing works focus on the model architecture or problem formulation, leaving the choice of transport cost relatively overlooked despite its potential for improvement.
>     For instance, the commonly used squared Euclidean distance assumes that samples lie in a Euclidean space, which is often not the case in practice. While frameworks like DeepJDOT address this by learning a latent space, we believe this approach is not always sufficient. Our proposed cost tackles this by employing the manifold assumption, using the Euclidean distance only locally.
>     Furthermore, our method considers intra-domain geometries and integrates source label information directly into the cost. The advantages of our approach, outlined in the final paragraph of the introduction in detail, are supported by extensive experimental results.
>     We briefly discussed the limitations of competing methods like ETD and RWOT, which also propose new transport costs, in the related work section.
>     As for other OT-based methods, we do not necessarily view them as competing approaches, as most of these methods can be applied using our proposed cost.
>
> 2. **Performance in the MNIST-USPS experiment:**
>
>     Since the code for RWOT is not available, we are unable to conduct a detailed analysis.
>     However, we observe that all baseline methods implemented using the DeepJDOT framework (DeepJDOT, JUMBOT, m-POT) achieved relatively low accuracy in this scenario, indicating that the DeepJDOT architecture may not be optimal for the MNIST-USPS experiment compared to the architecture used in RWOT. Additionally, we note that the proposed method outperformed all baselines in this scenario as well.
>
> 3. **The hyperparameter $\epsilon$:**
>
>     There is no single optimal methodology for selecting the Gaussian kernel parameter. However, a common practice is to set $\epsilon$ as the median of the pairwise distances in the data. This approach was utilized in all our deep experiments.

---

> ### Author Response · Authors · 2024-11-21
>
> This is comment 2 out of 2.
>
> 4. **Comparison to state-of-the-art methods:**
>     In our comparison, we included competing approaches that introduce new transportation costs for the OT problem and focus on the DA task &mdash; specifically, ETD and RWOT. Both methods define the cost as a weighted Euclidean distance, with the weights learned using different strategies.
>     For our method, we chose a simple framework &mdash; DeepJDOT &mdash; based on the ResNet50 model. We included comparisons with baselines that use the same framework but rely on the traditional OT cost, specifically the squared Euclidean distance.
>
>     We would like to point out that the research on DA is broad and ongoing. Since it is not feasible to compare with all the latest DA methods, we have chosen to focus exclusively on OT-based approaches. For this reason, we have opted not to include papers [2] and [5] in our comparison tables.
>     Regarding the recent method SPA [2], while it is not OT-based, it is a graph-based approach that also considers intra-domain structures, which makes it understandable why you find it relevant for comparison. However, since SPA relies on an adversarial training framework and includes additional objectives beyond the proposed graph spectral alignment term (such as adversarial loss and pseudo-labels loss), comparing it to our cost would not provide a fair evaluation.
>     Nonetheless, we acknowledge your valuable point.  While our current results using the DeepJDOT framework demonstrate how Diffusion-OT improves upon baselines and outperforms competing methods like ETD and RWOT, we agree that including additional OT-based methods, even those that do not propose a transportation cost,  in the literature review and comparison could further strengthen our evaluation.
>
>     It is worth highlighting that our cost is derived directly from the data and can be efficiently computed, and therefore it can be integrated into most OT-based methods.
>     The proposed method in [1], termed PPOT, is one such example; it introduces a regularization term in the optimization problem to encourage the learning of a representation space with better-separated classes. Although the method uses the squared Euclidean distance as the transport cost in the paper, it is not restricted to this choice, and our proposed cost can be easily incorporated into the framework.
>     Unfortunately, the code for PPOT has not been published. We have reached out to the authors to request access and are awaiting their response. We hope to provide results of Diffusion-OT within the PPOT framework by the end of the discussion period.
>
>     In [3], the squared Euclidean cost is used between clustering representations rather than between individual samples, making it incompatible with our proposed cost.
>     The paper presents two sets of results for each dataset: one using a DANN model with ResNet50 as the backbone for Office-Home and ResNet101 for VisDA, and the other using CDTrans, a SOTA transformer-based method.
>     When conducting a fair comparison against the ResNet-based model, Diffusion-OT achieves better results for Office-Home by an average of $0.7$% and outperforms in 8 out of the 12 scenarios.
>     While their method achieves $81.3$% with the CDTrans framework, the original CDTrans paper reports $80.5$% for this dataset, indicating that the improvement is attributable to the architecture rather than the method itself.
>     For VisDA, our method used ResNet50, whereas they used ResNet101, making a direct comparison unfair even with their lower result.
>     In [4], the authors propose an attention-based architecture for UDA and highlight its strong connection to the OT problem.
>     While we achieve significantly better results on the Office-Home dataset, the comparison is not fair due to the architectural differences and the use of different backbones for VisDA.
>
>     We acknowledge that our choice of architecture may limit the fairness of comparisons with SOTA methods.
>     To address this, we plan to evaluate our method with additional backbones and integrate it into a SOTA framework, such as CDTrans.
>     While obtaining results on a new architecture within the discussion period may not be feasible, we will make every effort to include them in the camera-ready version.

---

> > ### Comment · Reviewer_k4Z3 · 2024-12-03
> >
> > Thanks for the authors' responses. The discussions on hyperparameter settings and comparison methods address some of my concerns and have improved my opinion of the paper. However, since the paper only evaluates the method on ResNet-50, I believe the scope of evaluations remains too limited for a top conference like ICLR. Overall, I will raise my score to 5.

---

### Official Review · Reviewer_we8S · 2024-11-03

**Soundness:** 2
**Presentation:** 2
**Contribution:** 2
**Rating:** 6
**Confidence:** 3

**Summary:**

A novel transport cost, Diffusion-OT, is proposed in this paper for OT problem. By utilizing concepts of diffusion geometry, the authors derive an operator to quantify the probability of transporting between source and target samples. The authors give proof that the cost function is defined by an anisotropic diffusion process between the domains. Experiments show the superior performance of the proposed cost.

**Strengths:**

A new transportation cost, Diffusion-OT, is proposed which enables the learning of the geometries and relationships both between and within the two domains by considering both inter-domain distances and intra-domain structures.
By incorporating source label information into the cost, the proposed method is compatible with any OT solver and problem formulation.
Experiments demonstrate the effectiveness of the proposed method.

**Weaknesses:**

The results in Tab.1 and Tab.2 do not include all baseline methods, for example, results of  RWOT and ETD did not appear in the VisDA experiment.
The improvement on the digits dataset is relatively small.

**Questions:**

Does the proposed method has generalization on more general Universal Domain Adaptation tasks？Cause the Universal Domain Adaptation setting is more widely existing in practice.

---

> ### Author Response · Authors · 2024-11-21
>
> We thank the reviewer for their review and the helpful feedback.
> 1. **Missing results for competing methods:**
>
>     For ETD and RWOT, we reported the results as stated in their original papers. ETD did not provide results for the VisDA dataset. Although RWOT presented results for this dataset, the reported accuracy for DeepJDOT ($77.4$%) indicates that the ResNet model was modified, as it differs from both our reproduction ($69.85$%) and the original DeepJDOT paper ($66.9$%). Since the authors did not release the code or provide details of these modifications, we cannot determine what changes were made. Consequently, to ensure fairness, we chose not to include their results.
>
> 2. **Improvement on the Digits dataset:**
>
>     The improvement on the Digits dataset is indeed small. As mentioned in the paper, this dataset is considered an easier DA task, and the feature extractor appears to learn well-separated features for the classes, even when using the standard cost. Therefore, the minor improvement observed is expected.
>
> 3. **Generalization to Diffusion-OT:**
>
>     We thank you for the question and agree with your general observation. While this paper focuses on Unsupervised DA for coherence, we believe Diffusion-OT has the potential to be extended to other applications, which we consider as directions for future work.
>     In the meantime, one of the key advantages of our proposed cost is its adaptability to a broad range of OT-based methods, including those applicable to certain Universal Domain Adaptation tasks.

---

### Official Review · Reviewer_e86L · 2024-11-04

**Soundness:** 2
**Presentation:** 3
**Contribution:** 2
**Rating:** 3
**Confidence:** 5

**Summary:**

This paper proposes a domain adaptation method based on optimal transport. A diffusion optimal transport model is leveraged to construct a transport cost function between samples, in which intra-domain local geometry is introduced. The experiments are conducted on simulated and benchmark datasets to evaluate the performance of the proposed method.

**Strengths:**

1. In general, the paper is well-organized and easy to follow.

2. Both synthetic and real-world datasets are used in the experiments.

**Weaknesses:**

1. It is not new to introduce intra-domain geometry for domain adaptation. Intra-domain geometry has been widely considered in optimal transport and domain adaptation. For example, the Gromov-Wasserstein discrepancy considers transport between two metric spaces, and intra-domain geometry is involved in the construction of the metric. This has been introduced for cross-domain applications, as shown in [a][b].

2. Section 1 states that the weights in ETD and RWOT are learned, rather than directly derived from the data as in Diffusion-OT, which limits their applicability to deep learning. This statement is questionable. Intuitively, it is usually a good strategy to learn some properties such as weights or distance from data, which can adaptively extract geometric information involved in data and enhance the performance. Different from the learning strategy, the proposed method adopts a pre-defined approach to obtain a transport cost. The pre-defined paradigm may not obtain good performance if the adopted approach is not appropriate for real-world data.

3. Section 4 states that unsupervised domain adaptation implies that both domains are supported on the same hidden manifold. This assumption is vague. Domain shift could come from different marginal distributions, different conditional distributions, or some other factor. What is the specific assumption adopted in the submission? A detailed discussion should be provided.

4. The compared methods are out-of-the-date. Domain adaptation is an active area with many advances in recent years. It is easy to find state-of-the-art methods published recently as the comparison, such as (but not limited to) [c][d], which are also optimal transport-based methods for domain adaptation.

5. The results on Office-Home are lower than the results shown in [c]. Does the difference come from a different backbone model? If so, it is encouraging to adopt a better backbone to evaluate the performance of the methods.

6. It would be better to evaluate the impacts of the hyper-parameters $\epsilon$ used in the Gaussian kernel functions.

[a] Gromov-Wasserstein Learning for Graph Matching and Node Embedding, ICML 2019.

[b] Semi-Supervised Optimal Transport for Heterogeneous Domain Adaptation, IJCAI 2018.

[c] Probability-Polarized Optimal Transport for Unsupervised Domain Adaptation, AAAI 2024.

[d] COT: Unsupervised Domain Adaptation with Clustering and Optimal Transport, CVPR 2023.

**Questions:**

1. Section 4 states that unsupervised domain adaptation implies that both domains are supported on the same hidden manifold. This assumption is vague. What is the specific assumption adopted in the submission? A detailed discussion should be provided.

2. It would be better to conduct more state-of-the-art methods.

3. The results on Office-Home are lower than the results shown in [a]. Does the difference come from a different backbone model? If so, it is encouraging to adopt a better backbone to evaluate the performance of the methods.

4. It would be better to evaluate the impacts of the hyper-parameters $\epsilon$ used in the Gaussian kernel functions.

---

> ### Author Response · Authors · 2024-11-21
>
> This is comment 1 out of 2.
>
> We thank the reviewer for the thorough review and the valuable feedback.
> 1. **Intra-domain geometry using the Gromov-Wasserstein discrepancy:**
>
>     Typically, methods based on the Gromov-Wasserstein (GW) metric incorporate intra-domain geometry but do not account for inter-domain relationships.
>
>     In [a], the authors propose a method that learns node embeddings for both sets within the same metric space and computes the transport plan by solving an optimization problem that combines the GW discrepancy with cross-domain distances.
>     While this method does not specifically address DA applications, we recognize its potential applicability to certain DA tasks.
>
>     In [b], the authors present a method for Heterogeneous DA. The method includes adding a regularization term, $\Omega_l$, which contains inter-domain relationships.
>     However, the cross-domain distances are computed only **after** transportation, in the target domain, and require target labels, making this a semi-supervised approach.
>
>     We appreciate bringing these works to our attention and have included them in our paper (see paragraph 3 in the related work section, and paragraph 3 in Appendix A).
>
> 2. **Advantages of pre-defined transport costs over learned strategies:**
>
>     Although learning the cost is often an effective strategy, it is not always feasible. Our approach has the distinct advantage that the cost is not learned, making the method inherently compatible with both classical and deep learning frameworks. This flexibility ensures adaptability to a wide range of OT-based methods, including various problem formulations, efficient solvers, and architectures.
>     As demonstrated in the experiments section with the DeepJDOT framework, our method achieves superior performance in most scenarios compared to ETD and RWOT, particularly for real-world data.
>
> 3. **The manifold assumption:**
>
>     The assumption that both domains are supported on the same hidden manifold is central to our method, but it does not restrict the method's applicability to different domain shifts.
>     When using the squared Euclidean distance as the transport cost, whether explicitly stated or not, it is assumed that the source and target domains are embedded in $\mathbb{R}^d$.
>     The manifold assumption extends this concept, allowing for the possibility that the samples may not lie within a Euclidean space.
>     This assumption strengthens our approach to focus on the local relationships within and between domains, which we achieve using Gaussian kernels and the diffusion process.
>     The main limitation of this assumption is that both domains are required to share the same feature space, and it is common to most DA methods.
>     We have revised the beginning of Section 4 to provide a clearer explanation of this assumption.

---

> ### Author Response · Authors · 2024-11-21
>
> This is comment 2 out of 2.
>
> 4. **Comparison to state-of-the-art methods:**
>
>     In the experiments section, we compared our method with competing approaches that introduce new transportation costs for the OT problem and focus on the DA task &mdash; specifically, ETD and RWOT. Both methods define the cost as a weighted Euclidean distance, with the weights learned using different strategies.
>     For our method, we chose a simple framework &mdash; DeepJDOT &mdash; based on the ResNet50 model. We included comparisons with baselines that use the same framework but rely on the traditional OT cost, specifically the squared Euclidean distance.
>     Nonetheless, we acknowledge your valuable point. While our current results using the DeepJDOT framework demonstrate how Diffusion-OT improves upon baselines and outperforms competing methods like ETD and RWOT, we agree that using a SOTA architecture and comparing against SOTA OT-based methods, even those that do not propose a transportation cost, could further strengthen our evaluation.
>
>     It is worth highlighting that our cost is derived directly from the data and can be efficiently computed, and therefore it can be integrated into most OT-based methods.
>     The proposed method in [c], termed PPOT, is one such example; it introduces a regularization term in the optimization problem to encourage the learning of a representation space with better-separated classes. Although the method uses the squared Euclidean distance as the transport cost in the paper, it is not restricted to this choice, and our proposed cost can be easily incorporated into the framework.
>     Unfortunately, the code for PPOT has not been published. We have reached out to the authors to request access and are awaiting their response. We hope to provide results of Diffusion-OT within the PPOT framework by the end of the discussion period.
>
>     In [d], the squared Euclidean cost is used between clustering representations rather than between individual samples, making it incompatible with our proposed cost.
>     The paper presents two sets of results for each dataset: one using a DANN model with ResNet50 as the backbone for Office-Home and ResNet101 for VisDA, and the other using CDTrans, a SOTA transformer-based method.
>     When conducting a fair comparison against the ResNet-based model, Diffusion-OT achieves better results for Office-Home by an average of $0.7$% and outperforms in 8 out of the 12 scenarios.
>     While their method achieves $81.3$% with the CDTrans framework, the original CDTrans paper reports $80.5%$% for this dataset, indicating that the improvement is attributable to the architecture rather than the method itself.
>     For VisDA, our method used ResNet50, whereas they used ResNet101, making a direct comparison unfair even with their lower result.
>
>     We acknowledge that our choice of architecture may limit the fairness of comparisons with SOTA methods.
>     To address this, we plan to evaluate our method with additional backbones and integrate it into a SOTA framework, such as CDTrans.
>     While obtaining results on a new architecture within the discussion period may not be feasible, we will make every effort to include them in the camera-ready version.
>
> 5. **The hyperparameter $\epsilon$:**
>
>     There is no single optimal methodology for selecting the Gaussian kernel parameter. However, a common practice is to set $\epsilon$ as the median of the pairwise distances in the data. This approach was utilized in all our deep experiments.

---

> > ### Comment · Reviewer_e86L · 2024-12-02
> >
> > Thank the authors for the responses. However, several major concerns have not been well addressed.
> >
> > 1. The submission highlights the contribution of considering intra-domain geometry. However, intra-domain geometry has been well-studied in the Gromov-Wasserstein model. This highly weakens the novelty and the contribution of the submission.
> >
> > 2. It is difficult to figure out the advantages of pre-defined transport costs. Could the authors provide some situations where cost learning is not feasible?
> >
> > 3. The compared methods are out-of-the-date, and the comparison does not present state-of-the-art performance.
> >
> > 4. At least, the empirical results of different values of $\epsilon$ can be provided.

---

> > > ### Author Response · Authors · 2024-12-02
> > >
> > > We thank the reviewer for taking the time to review our responses and for clarifying their main concerns.
> > > 1. The Gromov-Wasserstein problem indeed incorporates intra-domain geometry as a generalization of the OT problem. Additionally, intra-domain geometry has been addressed in other frameworks, such as through regularization terms (e.g., Laplace regularization, as discussed in our related work and experiments).
> > > However, our contribution lies in proposing a new transport cost that integrates intra-domain geometry alongside inter-domain relationships and label information, all within a probabilistic framework.
> > > Therefore, we do not agree that the existence of other frameworks addressing intra-domain geometry weakens the novelty of our work. In fact, our proposed cost can be easily integrated into most of these frameworks, further enhancing their performance.
> > >
> > > 2. Beyond the obvious cases of small datasets or limited resources, certain tasks (e.g., medical imaging) benefit from domain knowledge that deep models might struggle to capture. For example, in Section 5.3, we demonstrated how a Riemannian framework can be leveraged for BCI datasets. The costs used in ETD and RWOT may not be easily adaptable to such a framework.
> > > We acknowledge that in most cases, a learned cost is both applicable and effective. However, while ETD and RWOT focus on learning a weighted Euclidean distance as the cost, our approach utilizes a diffusion process, which likely explains the superior performance of Diffusion-OT in the experimental results.
> > >
> > > 3. First, we note that since we propose a new transport cost, our primary competing methods are those that also introduce cost functions, such as ETD and RWOT. However, given the limited number of methods proposing alternative transport costs, we agree that a comparison with OT-based domain adaptation methods in general is necessary.
> > > That said, since our contribution is not a complete framework or architecture, and our goal is to fairly evaluate the performance of our transport cost against the standard OT cost (as well as the two competing methods), we chose not to include methods with significantly more powerful architectures than the one we use.
> > > Nevertheless, we appreciate your feedback and agree that we should compare to SOTA methods. We are currently working on integrating our cost into a SOTA architecture and will include comparisons with methods that have comparable architectures.
> > > We note again that our cost can be integrated into most OT-based methods, so we view these methods not as direct competitors, but as frameworks that our cost can enhance.
> > > Unfortunately, we are still awaiting responses from several authors regarding code or the ability to reproduce their results.
> > > As a result, we are unable to provide Diffusion-OT results with a SOTA architecture at this time. However, we are making every effort to ensure this is ready for the camera-ready version.
> > >
> > > 4. In general, the Gaussian kernel is highly sensitive to the scale parameter. If the scale is too large, the kernel will fail to capture important distinctions between the samples, resulting in a loss of relevant information. Conversely, if the scale is too small, the kernel may become overly sensitive to noise, leading to overfitting.
> > > Additionally, the scale parameter is dependent on the data scale, which is why a common practice is to choose the median of the data, as mentioned above.
> > > While the effect of $\epsilon$ on the proposed cost is similar to the effect of the scale parameter in any method that uses the Gaussian kernel for similarity measurement, we will consider adding an ablation study to the appendices to provide more intuition.

---

### Official Review · Reviewer_2EmH · 2024-11-04

**Soundness:** 3
**Presentation:** 3
**Contribution:** 3
**Rating:** 6
**Confidence:** 4

**Summary:**

This paper proposes a novel transportation cost function, termed Diffusion-OT, for the Optimal Transport (OT) problem in the context of domain adaptation. Diffusion-OT leverages concepts from diffusion geometry and manifold learning to account for both intra-domain and inter-domain relationships. The proposed cost function is derived from a composite diffusion operator that consists of three diffusion steps: within the source domain, across domains, and within the target domain. By incorporating source label information into the diffusion process, Diffusion-OT can guide the anisotropic diffusion according to class labels. Experiments on various benchmarks demonstrate that Diffusion-OT outperforms competing methods, achieving state-of-the-art results on non-Euclidean data.

**Strengths:**

1. The proposed Diffusion-OT cost function is a novel approach that goes beyond the traditional squared Euclidean distance used in OT for domain adaptation. It considers both intra-domain and inter-domain geometries.
2. Experimental results show that Diffusion-OT achieves superior performance compared to baseline and recent OT-based methods across multiple datasets, demonstrating its effectiveness in domain adaptation tasks.

**Weaknesses:**

1. The first concern is the complexity and computational cost. The composite diffusion operator involves multiple steps and may lead to higher computational cost compared to simpler cost functions. The computational complexity of the proposed method, especially when dealing with large-scale datasets, is not fully discussed.

2. Theoretical analysis limitations: While the paper provides theoretical analysis, it mainly focuses on the asymptotic behavior of the diffusion operators. A more rigorous analysis of the convergence properties and error bounds of the proposed method would strengthen the theoretical foundations.

3. Incomplete analysis of failure cases: Although the authors admit the limitation of the proposed method is its assumption that both source and target domains reside in the same space. The authors do not provide detailed analysis of failure cases. As a result, we cannot clearly evaluate the negative impact when applying the proposed method in the real-world applications where the domains have substantially different underlying structures. This could limit the generalizability of the proposed method.

**Questions:**

See weaknesses.

---

> ### Author Response · Authors · 2024-11-21
>
> We thank the reviewer for the thorough review and the valuable feedback.
> 1. **Computational complexity:**
>
>     Thank you for raising this point.
>     In response, we added a discussion on the method's computational complexity in Appendix C.2, and we provide it here as well.
>     The most commonly used transport cost, the pairwise squared Euclidean distances, has a computational complexity of $\mathcal{O}(n^2)$, assuming balanced datasets.
>     In contrast, our proposed cost involves the following steps:
>     1. Computing pairwise distances for three matrices:  $\mathcal{O}(n^2)$.
>     2. Applying the exponential function to the matrices: $\mathcal{O}(n^2)$.
>     3. Normalizing the matrices to be row-stochastic: $\mathcal{O}(n^2)$.
>     4. Multiplying the three probability matrices: $\mathcal{O}(n^3)$.
>     5. Applying the logarithm to the final diffusion operator: $\mathcal{O}(n^2)$.
>
>     As noted in Appendix D, we often use doubly-stochastic normalization instead of row-stochastic normalization, employing the Sinkhorn algorithm. Since the complexity of Sinkhorn is also $\mathcal{O}(n^2)$, this does not increase the overall complexity.
>     In total, the computational complexity of the proposed cost is $\mathcal{O}(n^3)$.
>     While higher than the traditional cost, our method avoids the need for a regularization term, which can often add to the optimization complexity.
>
>     Additionally, we note that up to the fourth step, the computation involves three $n \times n$ matrices rather than one, resulting in higher memory requirements.
> 2. **Theoretical results:**
>
>     We agree that such theoretical results could strengthen the theoretical foundations. Unfortunately, we do not have these results at present.
>     We would like to note that none of the competing methods offer theoretical results. In contrast, while our analysis focuses on the asymptotic behavior of the proposed diffusion operator, we provide a detailed analysis that motivates our approach. Furthermore, to establish confidence in the superiority of our method over traditional approaches, we conducted extensive experiments across two distinct frameworks and multiple datasets.
> 3. **Failure cases:**
>
>     Thank you for the comment.
>     First, we would like to emphasize that our assumption that both domains reside in the same space does not present a barrier when the underlying structures differ.
>     When using the squared Euclidean distance as cost, whether explicitly stated or not, it is assumed that the source and target domains are embedded in $\mathbb{R}^d$.
>     The manifold assumption extends this concept, allowing for the possibility that the samples may not lie within a Euclidean space.
>     Notably, the manifold assumption does not compromise performance for the UDA tasks presented. However, our method is currently limited to scenarios where the source and target domains share the same feature space (specifically, due to the second diffusion step involving inter-domain distances). All competing methods have this limitation as well.
>     We have revised the beginning of Section 4 to provide a clearer explanation of this assumption.
>
>     In our empirical experiments on real-world data, as well as in our simulations, we did not observe any clear failure cases. However, it is important to note that our proposed cost function contains a hyperparameter, and the method's performance may vary depending on its choice.

---

### Meta-Review · Area_Chair_4WmT · 2024-12-15

**Metareview:**

This paper introduces diffusion optimal transportation as a method for domain adaptation and evaluates its effectiveness on several small benchmarks. However, it has received mixed reviews, with a generally negative inclination. As noted by Reviewer e86L, the proposed model lacks novelty, as intra-domain analysis has been previously explored in the literature. Additionally, the paper's performance does not achieve state-of-the-art (SoTA) results on key benchmarks such as Office-Home. Furthermore, the work would benefit from a broader range of comparisons in the domain adaptation (DA) field to better position its contributions.

**Additional Comments On Reviewer Discussion:**

The evaluation is not enough such as the backbone of ResNet is used, the benchmarks are small-scale.

---

### Decision · Program_Chairs · 2025-01-22

Reject